



# Benthic foraminifera as tracers of brine production in Storfjorden "sea ice factory"

Eleonora Fossile[1], Maria Pia Nardelli[1], Arbia Jouini[1], Bruno Lansard[2], Antonio Pusceddu[3], Davide Moccia[3], Elisabeth Michel[2], Olivier Péron[4], Hélène Howa[1] & Meryem Mojtahid[1]

[1] LPG-BIAF UMR 6112, Univerité d'Angers, CNRS, UFR Sciences, 2 bd Lavoisier 49045, Angers Cedex 01, France
[2] LSCE, UMR 8212, IPSL-CEA-CNRS-UVSQ-Université Paris Saclay, 91198 Gif-sur-Yvette, France
[3] Department of Life and Environmental Sciences, University of Cagliari, 09126 Cagliari, Italy
[4] SUBATECH, UMR 6457, CNRS-Université de Nantes, 4 rue A. Kastler, 44307 Nantes, France

*Correspondence to*: Eleonora Fossile (eleonora.fossile@etud.univ-angers.fr)

**Abstract.** The rapid response of benthic foraminifera to environmental factors (e.g., organic matter quality and quantity, salinity, pH) and their high fossilisation potential make them promising bio-indicators for the intensity and recurrence of brine formation in Arctic seas. Such approach, however, requires a thorough knowledge of their modern ecology in such extreme settings. To this aim, seven stations along a N-S transect across the Storfjorden (Svalbard archipelago) have been sampled using an interface multicorer. This fjord is an area of intense sea ice formation characterised by the production of Brine-

enriched Shelf Waters (BSW) as a result of a recurrent latent-heat polynya. Living (Rose Bengal stained) foraminiferal assemblages were analysed together with geochemical and sedimentological parameters in the top five centimetres of the sediment. Three major biozones were distinguished: i) the "inner fjord" dominated by typical glacier proximal calcareous species which opportunistically respond to fresh organic matter inputs; ii) the "deep basins and sill" characterised by glacier distal agglutinated faunas. These latter are either dominant because of the mostly refractory nature of organic matter and/or

the brine persistence that hampers the growth of calcareous species and/or causes their dissolution. iii) The "outer fjord" characterised by typical North Atlantic species due to the intrusion of the North Atlantic water in the Storfjordrenna. The stressful conditions present in the "deep basins and sill" (i.e. acidic waters and low food quality) result in a high agglutinated/calcareous ratio (A/C). This supports the potential use of the A/C ratio as a proxy for brine persistence and overflow in Storfjorden.

## 25 1 Introduction

The polar regions are particularly sensitive to climate change as evidenced from the several dramatic alterations in recent decades (Peings, 2018). For instance, Arctic surface temperatures have increased at twice the global rate (i.e., Arctic amplification; Holland and Bitz, 2003; Dai et al., 2019) while sea ice cover has been steadily decreasing over recent decades both in extent and volume (IPCC, 2013; Labe et al., 2018). With less ice present, the ocean surface absorbs considerably more

sunlight energy. This leads to further warming of the atmosphere and the ocean, therefore enhancing sea ice melting, which, in turn, affects marine and continental ecosystems (Perovich and Richter-Menge, 2009).





The short period of historical and satellite observations (~100 yrs) only opens a narrow window on the natural variability of sea ice dynamics. In this context, the uncertainties are larger than for direct measurements, and longer time series are increasingly needed to place the recent trends in a longer-term perspective (i.e., multi-centennial time-scale) (Nicolle et al.,
2018). A recent review study compiling several high-resolution terrestrial proxies show that the modern decline in Arctic sea ice is unprecedented since at least the last few thousand years and unexplainable by known natural variability (e.g., Kinnard et al., 2011). To better understand how and how much natural and anthropogenic forcing factors control the sea-ice dynamics, there is a need for more high-resolution marine time-series covering the historical period, and for robust biological proxies in key areas from the circum-Arctic. Storfjorden, a semi-enclosed bay located in the Svalbard archipelago (Fig.1a), is one of the
Arctic regions particularly suitable for studying first-year sea ice dynamics. Indeed, Skogseth et al. (2004) defined Storfjorden as an "ice factory" because of the presence of a recurrent coastal polynya that contributes to about 5-10% to the total brine waters produced on Arctic shelves (Smedsrud et al., 2006). Brines are salty and $CO_2$-rich waters (i.e. low pH) (Rysgaard et al., 2011) that are produced when sea-ice forms in inner fjords. Because of their high density, they cascade after mixing with shelf waters (Skogseth et al., 2005a), and ventilate the deep sea (Rumohr et al., 2001). During cascading, brines may cause
sedimentary disturbance as they can release down-slope turbidity flows (Rumohr et al., 2001) and, at the meantime, export particulate and dissolved inorganic and organic carbon from the productive surface waters down to the seafloor (Anderson et al., 2004).

Benthic foraminifera are among the most abundant protists living in marine ecosystems, from brackish waters to abyssal plains (e.g., Murray, 2006). Due to their short life cycle, high diversity and specific ecological requirements, they respond quickly to
the physicochemical environmental conditions (e.g., organic inputs, oxygenation, pH) by increasing their density (e.g., Kitazato et al., 2000) changing the faunal composition or modifying their microhabitat (Jorissen et al., 1995; Ohga and Kitazato, 1997). Therefore, benthic foraminifera are potentially good proxies directly for brine waters that bathe the deep-sea arctic environments, and, indirectly, for sea-ice production. The existing benthic foraminiferal-based studies in Storfjorden used fossil faunas to interpret late Quaternary paleoenvironmental changes (Rasmussen and Thomsen, 2009, 2014, 2015).
Although highlighting major difficulties in the interpretation of most of these paleorecords without modern proxy calibrations, those studies further suggested the ratio of agglutinated to calcareous benthic foraminifera as a potential proxy for brine changes. Living foraminiferal distributions have been, however, studied in several Svalbard fjords, in particular in relation to the distance from continental glaciers and the associated sedimentary supply (Hansen and Knudsen, 1995; Korsun et al., 1995; Hald and Korsun, 1997; Korsun and Hald, 2000; Sabbatini et al., 2007; Ivanova et al., 2008; Forwick et al., 2010; Zajaczkowski
et al., 2010; Łącka & Zajączkowski, 2016; Jernas et al., 2018). To our knowledge, none of these studies targeted the influence of Brine-enriched Shelf Waters (BSW).

To develop a robust biological marine proxy of environmental variation based on communities of benthic fossil foraminifera, it is beforehand crucial to well constrain their modern ecology in this specific environment. To provide new insights on this issue, here we investigate living (rose Bengal stained) benthic foraminiferal faunas from Storfjorden and their response to
various measured environmental variables (e.g. sediment characteristics, organic matter quantity and composition, sediment



oxygen and pH micro-profiles) that are further linked with BSW. Furthermore, the interpretation of our results integrates the extended literature of the physical oceanography in this specific semi-enclosed bay (e.g., Haarpaintner et al., 2001a, 2001b; Omar et al., 2005; Skogseth et al., 2005a, 2005b, 2008; Geyer et al., 2009; Jardon et al., 2014).

## 2 Oceanographic and environmental settings

The Svalbard archipelago, located north of the Arctic circle, extends from 74° N to 81° N and 10° E to 35° E (Fig. 1a). It is surrounded by the Arctic Ocean to the north, the Barents Sea to the south and east, and the Norwegian-Greenland Sea to the west. Storfjorden, the biggest fjord in the Svalbard archipelago, is approximately 190 km long with a maximum water depth of ~190 m present in a central deep glacial trough (referred to as deep basins in Fig. 1b). The northern Storfjorden (i.e. inner fjord; Fig. 1b) is connected with the northwestern Barents Sea by two sounds (Heleysundet and Freemansundet) through where

relatively energetic tidal exchanges occur (McPhee et al., 2013). To the south, a sill (77°N-19°E) about 120 m deep separates the inner Storfjorden and the deep basins from the outer Storfjorden trough (Storfjordrenna) (Fig. 1a), a 200-300 m deep glacial paleo-valley that incised the western Barents Sea continental margin during previous sea level low-stands (Pedrosa et al., 2011).

The Svalbard archipelago is influenced by two major water masses. Along the eastern and southern margin of Svalbard, cold

and relatively low saline Arctic waters flow out from the Barents Sea via the East Spitsbergen Current (ESC) (Fig. 1a). In the eastern Norwegian-Greenland Sea, the main stream of Atlantic Water which is the most important source of heat and salty water into the Arctic Ocean, is carried northwards by the Norwegian Atlantic Current (NAC) (Fig. 1a). North of Norway, the NAC splits into two branches: i) the Norwegian Current (NC, or Norwegian Coastal Current) that enters the Barents Sea eastward around 70°N (not shown in Fig.1a) along the northern coast of Norway, and ii) the West Spitsbergen Current (WSC)

that flows northwards along the western Svalbard coast towards the Fram Strait (Schauer, 1995). Recent studies report fluctuations in heat transport to the Arctic Ocean by the WSC in particular in link with global climate changes (e.g. Holliday et al., 2008; Piechura and Walczowski 2009; Beszczynska-Moller et al., 2012). This current is playing a significant role in the process of recent Arctic warming, by influencing the sea-ice distribution and cover in Svalbard (Haarpaintner et al., 2001; Polyakov et al., 2012).

The water masses in Storfjorden have two main origins: warm Atlantic waters and cold Arctic waters. These are mostly separated by the location of the polar front that shifts seasonally and conditions therefore the northward or southward position of these water masses (Loeng, 1991) (Fig. 1a). The warm Atlantic surface waters carried by the NAC enter the Storfjordrenna from the southwest (Wekerle et al., 2016) (Fig. 1a). During spring-summer, this latter flows into Storfjorden along its eastern margin following a cyclonic circulation (Nielsen and Rasmussen, 2018; Piechura and Walczowski, 2009). The cold Arctic

waters derived from the ESC enter Storfjorden from the east through narrow topographic gateways (Heleysundet and Freemansundet sounds), and the topographic depression north of Storfjordbanken (Fig. 1a). This Arctic water circulates cyclonically through the fjord, flowing southwards along the western Storfjorden coast and continues northwards as a coastal





current along with the west Spitsbergen coast (Nielsen and Rasmussen, 2018; Rasmussen and Thomsen, 2015). Vertically, water masses are usually arranged in three main layers within an Arctic fjord with a sill (Farmer and Freeland 1983): a relatively fresh surface layer, a deep and saline layer below the sill depth, and an intermediate layer in between (Fig. 1b). Profiles from late summer in Storfjorden show a well-mixed fresh surface layer extending down to 40 m depth separated from the intermediate layer (comprising advected Atlantic Water) by a steep halocline. The deepest layer, which sits below the sill depth, is a cold and saline water mass derived from trapped brines (e.g., Skogseth et al., 2005a; Cottier et al., 2010; Rasmussen and Thomsen, 2015).

The shelf sea in the Storfjorden is characterised by an extended winter sea-ice cover due to the presence of a recurrent winter coastal latent-heat polynya mostly located in the northeast part (Skogseth et al., 2004). These are ice-free areas formed and maintained by advection of ice by off-shore winds, tidal and ocean currents. The opening of a latent-heat polynya determines an intensive heat loss to the atmosphere that can lead to a persistent ice formation (Fer, 2004; Skogseth et al., 2005a). Polynya particularly occur when north-easterly winds intensify in winter (Skogseth et al., 2005). The continuous production of thin, first-year sea ice, which generally starts in December (Smedsrud et al., 2006), leads to a subsequent formation of brine waters in Storfjorden. Brines are cold, dense and well oxygenated waters, enriched in salt, total dissolved inorganic carbon (DIC) (i.e. low pH) that are rejected in under-sea ice waters when sea ice is formed (Rysgaard et al., 2011; Anderson et al., 2004). The shelf convection promotes the mixing of brines with shelf waters, leading to the formation of Brine-enriched Shelf Waters (BSW). In the early winter freezing period, the extremely dense BSW sink, filling the deeper basins and pushing the less dense waters above the sill level causing a weak overflow (Skogseth et al. 2005a) (Fig.1b). During winter the low temperature causes a brine volume contraction and a decrease in the sea ice permeability that prevents the air-sea ice gas exchange; brine volume contraction causes a further increase of brine salinity and $CO_{2(aq)}$ (Rysgaard et al., 2011). The continuous freezing in spring causes the accumulation of BSW in the deep basins and a strong steady overflow period over the sill. Although weaker, the overflow continues even in summer after the end of the freezing period. At the meantime, the fresh melting surface water is warmed by surface heating (Skogseth et al., 2005). During spring and summer, the ice melting reduces $CO_{2(aq)}$ (Rysgaard et al., 2011) and the increase of light availability (Horner and Schrader, 1982) triggers ice algae photosynthetic activity which further reduces DIC concentrations of surface waters (Gleitz et al.,1995). In autumn, surface waters lose heat and become colder. At this time, the old BSW are trapped in the deep basins, but strong wind events cause occasional discharges over the sill (Skogseth et al., 2005a) (Fig. 1b). The entire Arctic coastal polynyas produce about 0.7-1.2 Sv (1 Sv = $10^6$ m$^3$/s) of BSW (Cavalieri and Martin, 1994) providing about 10% of the deep water formed in the Arctic Ocean and Barents Sea today (Smethie et al., 1986; Quadfasel et al., 1988; Rudels and Quadfasel, 1991). Storfjorden is a major supplier of BSW, producing alone 5-10% of the dense water in the Arctic Ocean (Quadfasel et al., 1988; Skogseth et al., 2004).

Winkelmann and Knies (2005) classified the inner Storfjorden as a low-energetic environment characterised by high sedimentation rates and organic-rich sediments (total organic carbon content (TOC) >2%) with high proportion of terrestrial component.



## 3 Material and Methods

### 3.1 Interface sediment sampling and CTD profiles

In July 2016, seven stations were sampled along a N-S transect in Storfjorden (Fig.1a, Table 1) during the STeP (Storfjorden
Polynya Multidisciplinary Study) cruise onboard the R/V *L'Atalante* (IFREMER). Stations MC1 to MC3 are positioned on
the continental shelf at the head of the fjord, stations MC4 and MC5 are located in the deep central basins, station MC6 is
located on the sill, and station MC7 is located in Storfjordrenna (Fig.1a, Table 1). At each station, 10 to 40 cm long sediment
cores were sampled using a multicorer (10 cm Ø) in order to get undisturbed sediment-water interfaces. Three replicate cores
were sampled at each station (except for station MC3 where only two cores were collected): the first core for geochemical
analysis (oxygen, pH and porosity profiles), the second one for $^{210}Pb_{xs}$ dating, grain size, phytopigment and organic matter
analyses, and the third one for foraminiferal analysis.

In order to determine the main environmental characteristics of each site, hydrographic casts were performed with a
Conductivity-Temperature-Depth (Seabird 911 plus CTD) equipped with a fluorometer. A rosette sampler supplied with
22*12-L Niskin bottles was used for water-column sampling. Bottles were fired at standard depths to measure oxygen,
nutrients and Chlorophyll-*a*.

### 3.2 Geochemical analyses

Immediately after the recovery of sediment cores, oxygen and pH microprofiles have been measured at the sediment-water
interface. We used a micromanipulator that can drive $O_2$ and pH microelectrodes (Unisense®) at the same time with a 200 µm
vertical resolution. Oxygen profiles were performed using Clark-type microelectrodes with a 100 µm thick tip (Revsbech
1989), while pH profiles were measured using a glass microelectrodes with a 200 µm thick tip. The $O_2$ concentration of bottom
water was analysed by Winkler Titration (Grasshoff et al. 1983). At each station, triplicate samples were analysed with a
reproducibility of ± 2 µmol L$^{-1}$. The pH microelectrodes were calibrated using NBS buffer solutions (pH 4, 7 and 10). The pH
of bottom water was also determined by spectrophotometry using mCresol Purple as dye (Dickson et al. 2007). All pH
measurements were recalculated at *in situ* temperature, salinity and depth using CO2SYS (Pierrot et al., 2006) and were
reported on the total proton scale ($pH_T$). The measurements for both $O_2$ and pH profiles, were repeated many times in order to
assess the reproducibility of the measurements and the natural heterogeneity of these parameters in the sediment.

### 3.3 $^{210}$Pb dating and grain size analysis

At each station (except for the MC3), one core was sliced on board collecting five sediment layers (0-0.5, 0.5-1, 1-2, 2-5 and
5-10 cm), then stored at -20°C. In the land-based laboratory, an aliquot of sediment was sampled for grain-size analyses and
the rest was lyophilised for the $^{210}Pb_{xs}$ analyses. Grain-size analyses were performed using the laser diffraction particle size
analyser Malvern Mastersizer 3000. The particle-size distributions were analysed with GRADISTAT 8.0 software program
(Blott and Pye, 2001). The replicated analyses were run for each sample aliquot and the most representative was selected. For





the analysis of faunas in response to environmental parameters, the grain size of the superficial sediment layer (0.0-0.5 cm depth) was considered as representative of the sediment-water interface characteristics. Another aliquot of sediment was freeze-dried for gamma spectrometry measurements in order to determine the apparent sedimentation rate by the $^{210}Pb_{xs}$ method (Appleby and Oldfield, 1978). $^{210}Pb$ dating have been conducted using a gamma spectrometer Canberra® HPGe GX4520 coaxial photon detector. The homogenised samples were weighed and sealed in a defined geometry for at least three weeks to ensure $^{222}Rn/^{226}Ra/^{214}Pb$ equilibration. Sedimentation rate was based on the determination of the excess or unsupported activity $^{210}Pb$ ($^{210}Pb_{xs}$) and performed through constant flux - constant sedimentation (CFCS) model (Sanchez-Cabeza and Ruiz-Fernández, 2012). $^{210}Pb_{xs}$, incorporated rapidly into the sediment from atmospheric fallout and water column scavenging was calculated as the difference between the total measured $^{210}Pb$ activity (supported + excess) at 46.54 keV and $^{214}Pb$ at 351.93 keV.

**3.4 Organic matter quantity and biochemical composition**

To assess the quantity and biochemical composition of the organic matter, the top half centimetre of the sediment cores was sliced on board and immediately stored at -20°C until analysis. As the redox fronts and foraminiferal microhabitats in the sediment are strictly driven by the organic matter supply at the sediment-water interface (e.g. Jorissen et al.,1995), only the organic matter data for the first upper half centimetre were used to interpret the faunal distribution.

In the laboratory, chlorophyll-*a*, phaeopigment, lipid, carbohydrate and protein contents were determined on three pseudo-replicates (ca. 1 g wet sediment). Chlorophyll-*a* and phaeopigment analyses were carried out according to Lorenzen and Jeffrey (1980). Briefly, pigments were extracted with 90% acetone (12 h in the dark at 4°C). After the extraction, the pigments were fluorometrically analysed to estimate the quantity of Chl-*a* and, after acidification (20 s) with 0.1 N HCl (Plante-Cuny, 1974), to estimate the amount of phaeopigments. Chloroplastic pigment equivalents (CPE) were calculated as sum of Chl-*a* and phaeopigment contents, and carbon associated with CPE (C-CPE) was calculated by converting CPE contents into carbon equivalents using a factor of 30μgC μg phytopigment$^{-1}$ (de Jonge, 1980). Protein, carbohydrate and lipid sedimentary contents were determined by spectrophotometry (Danovaro, 2009) and concentrations reported as bovine serum albumin, glucose and tripalmitin equivalents (mg per gram of dry weight sediment), respectively. Protein, carbohydrate and lipid concentrations were converted into carbon equivalents using the conversion factors 0.49, 0.40 and 0.75 g C g$^{-1}$, respectively (Fabiano et al., 1995). The sum of protein, carbohydrate and lipid carbon was referred to as biopolymeric Carbon (BPC; Tselepides et al., 2000) that represents the semi-labile fraction of the total organic carbon (Pusceddu et al., 2009; Van Oevelen et al., 2011). The algal fraction of biopolymeric C, proxy for the most labile fraction of sedimentary organic matter (Danovaro and Pusceddu, 2003; Pusceddu et al., 2010) was calculated as the percentage ratio of CCPE on BPC.

**3.5 Living foraminiferal fauna sampling and analyses**

Immediately after sampling, interface cores were sliced horizontally every 0.5 cm between 0 and 2 cm, every 1 cm from 2 down to 6 cm, and every 2 cm from 6 to 10 cm depth. Each slice was stored in a 500 cm$^3$ plastic bottle filled with 95% ethanol





containing 2 g L$^{-1}$ of Rose Bengal stain (in order to label living foraminifera). In the laboratory, sediment samples were sieved through 63, 125 and 150 µm meshes and the resulting fractions were stored in 95% ethanol. All living (Rose Bengal stained) specimens from the >150 µm fraction were hand-picked in water from the surface layer down to 5 cm depth. Additionally, the living foraminifera of the 63-150 µm fraction were picked only for the first centimetre of sediment, in order to investigate the potential use of this size fraction for ecological consideration.

Samples of the smallest size fraction, showing very high benthic foraminiferal abundance, were dried at 50°C and split with an Otto Microsplitter. Then foraminifera were hand-sorted from an entire split containing a minimum of 300 individuals and the counts were extrapolated for the total sample.  Foraminiferal biodiversity was estimated using different diversity indices: species richness (S) measured as the number of species, species diversity (H log$_e$) measured by the Shannon–Wiener (H') information function and species evenness (J) measured using the Pielou Index (1975). All indices were calculated using the

Paleontological Statistics Data Analysis (PAST) software (version 2.17c; Hammer et al., 2001). Foraminiferal densities are expressed per 50 cm$^2$ (when considering total densities) and per 50 cm$^3$ volume (when considering layers of different thickness). The agglutinated species *Spiroplectammina earlandi* and *Spiroplectammina biformis* were not distinguished because these are morphotypes of the same species according to Korsun and Hald (2000).

### 3.6 Multivariate analyses

A Canonical Correspondence Analysis (CCA) was used to investigate the relationships between the environmental parameters (depth, bottom water temperature, salinity, oxygen penetration depth (OPD), sediment porewater pH, sediment grain size and organic matter) and the faunas (>150 µm, 0-5 cm) of all stations considering only the absolute densities (ind. 50cm$^{-2}$) of the species which contribute with >5% to the assemblage. We used the grain size characteristics and the organic matter contents and composition of the uppermost sediment layer (0.0-0.5 cm). Values of different environmental variables and different orders

of magnitude were homogenised using the following standardisation: ($x$-mean $x$)/$sd$, in which $x$ is the value of the variable in one station, *mean x* is the mean of the same variable among the stations and *sd* is the corresponding standard deviation. Non-metric multidimensional scaling (nMDS) bi-plots and cluster analysis (Bray-Curtis similarity) were used to visualize the differences between stations and size fractions. The analyses were conducted on the foraminiferal assemblages of the topmost centimetre of sediment considering separately the smaller fraction 63-150 µm, the >150 µm fraction and the total assemblage

(>63 µm fraction). The densities of the foraminiferal faunas were normalised using the following transformation: Log$_{10}$($x$+1), where $x$ is the density expressed in ind. 50cm$^{-2}$ (considering the 0-5 cm sediment interval for the CCA and the 0-1 cm interval for the nMDS and cluster analysis). All these multivariate analyses were performed using the PAST software (version 2.17c; Hammer et al., 2001).

### 3.7 Visual characterisation of test dissolution

Using high-resolution SEM images of specimens from the >150 µm size fraction (Fig. S2), we qualitatively distinguish four dissolution stages from weak to severe, following the classification of Gonzales et al. (2017): stage I) no sign of dissolution,




transparent tests and smooth surfaces; stage II) whitish tests with visible pores, and frequently, the last chamber is lost as well as the first calcite layers; stage III) several chambers are dissolved and the remaining ones present opaque wall tests; stage IV) nearly complete dissolution of the tests and only the organic material remains. The percentages of specimens belonging to each of the four stages in all samples were not quantified because of the potential loss of information due to the bad preservation characterising the two most severe dissolution stages.

## 4 Results

### 4.1 Bottom water properties

In July 2016, bottom waters at the inner fjord stations MC1 and MC2 are cold (below -1.5°C) and relatively salty (34.89 and 34.79 respectively), while station MC3 presents a positive bottom water temperature (1.10°C), the lowest salinity (34.74) and the highest $pH_T$ (8.12) and $O_2$ (350 µmol.L$^{-1}$) of the fjord transect (Table 1). The two deep basin stations (MC4 and MC5) display the lowest bottom water temperature (both -1.78 °C), the lowest $pH_T$ (7.92 and 7.91, respectively) and the highest salinity (34.92 and 34.93, respectively) (Table 1). The sill station MC6 shows the same range of salinity than in the inner fjord (34.8) with a slightly higher temperature (-1.13°C). The outer fjord station MC7 records the highest temperature (3.53°C) and salinity (35.05) of the sampled transect. The shallowest stations (MC1 and MC3) are well ventilated with $O_2$ concentration higher than 340 µmol L$^{-1}$. Stations MC2, MC4, MC5 and MC6 show the same bottom water $O_2$ concentration (318 ± 2 µmol L$^{-1}$). The deepest station (MC7) located outside Storfjorden shows a lower $O_2$ concentration (305 µmol L$^{-1}$).

### 4.2 $^{210}$Pb dating and grain size analyses

The $^{210}$Pb age models show a relatively high sedimentation rate at all stations with an average of 3.6 ± 0.4 mm yr$^{-1}$ (see supplementary materials Table S1 for more details), except at the outer fjord station MC7 where sedimentation rate is much lower (1.3 ± 0.6 mm yr$^{-1}$; Table S1).

Grain size analyses of the topmost half centimetre of the sediment indicate the presence of fine silt sediments at all stations (Table S1). Slight differences are however noted in terms of the mode (approximately 10 µm in the fjord and 20 µm at the outer station MC7) and the percentage of sand which increases from approximately 4% at MC1 to 10.4% at station MC6.

### 4.3 Biogeochemical analyses of the sediment

The sediment oxygen profiles (Fig. S1a) at the inner fjord stations (MC1-MC3) display an average oxygen penetration depth (OPD) of 7.7 ± 1.0 mm (n=3), 4.9 ± 0.4 mm (n=4) and 4.8 ± 1.9 mm (n=6), respectively. The OPD at the deep basin stations (MC4 and MC5) and at the sill station (MC6), are 5.7 ± 1.1 mm (n=3); 6.2 ± 0.9 (n=10); 8.6 ± 3.8 (n=6), respectively. The outer fjord station (MC7) shows the highest OPD of the sampled transect (15.6 ± 1.0 mm).

In order to better highlight the differences in porewater pH between the stations, we conducted a one-way ANOVA on the $pH_T$ values found at the sediment-water interface (based on pH profiles; Fig. S1b). This analysis shows significant differences





among the stations (F= 128.8, p < 0.001). The inner fjord stations (MC1-MC3) present $pH_T$ values generally above 7.95 whereas the deep basin stations display values lower that 7.90 units (7.84 and 7.90 for MC4 and MC5 respectively). The $pH_T$ at the sill station (MC6) is 7.98, while at the outer fjord station (MC7) $pH_T$ is 8.00.Tukey HSD test shows that station MC3

presents a porewater $pH_T$ value significantly higher than all the other stations (p < 0.001), whereas the two deep basin stations MC4 and MC5 are characterised by the lowest porewater pH values (p < 0.001). When considering the entire profiles, pH strongly decreases in the topmost part of the sediment (0-5 mm) at all stations but with different slopes. In fact, the strongest pH gradient is observed at the MC3 station (-0.2 pH unit $mm^{-1}$). By contrast, the pH decrease at the outer fjord station MC7 is the slowest found in the transect (-0.1 pH unit $mm^{-1}$) (see Fig. S2b).

Concerning the organic matter, the results for BPC, PRT, CHO, CPE content and algal fraction of BPC (C-CPE/BPC) are presented in Fig. 2. The complete dataset is reported as average ± standard deviation (n = 3) in Table S2. The BPC (Fig. 2a) varies significantly among the stations (one-way ANOVA, F = 21.72, p < 0.001). Stations MC1 and MC7 have values of BPC significantly lower (5.49 ± 0.49 mgC $g^{-1}$ and 4.71 ± 0.07 mgC $g^{-1}$ respectively, p < 0.05) than at all other stations. In these latter, the average BPC varies between 6.86 ± 0.45 mgC $g^{-1}$ and 7.38 ± 0.21 mgC $g^{-1}$. The PRT contents (%) (Fig. 2a) of the

BPC varies significantly among the stations (One-way ANOVA, F = 6.94, p < 0.01). In particular the deep basins present significantly lower percentages of PRT compared to all the other stations (32.12 ± 4.42 % at MC4 and 30.75 ± 1.69 at MC5; p < 0.01). The CHO contents (%) (Fig. 2a) change significantly among the stations (One-way ANOVA, F = 46.6, p < 0.001), displaying the highest scores in the deep basins (33.79 ± 1.71 % and 36.08 ± 2.52 % at MC4 and MC5 respectively). The CPE (Fig. 2b) varies significantly among the stations (one-way ANOVA, F = 52.03, p < 0.001). CPE content is considerably lower

in the outer fjord station MC7 (6.43 ± 0.45 µg $g^{-1}$) compared to all other stations (p < 0.001). Inside the fjord, station MC1 differs from MC2 with values of 24.04 ± 3.69 µg $g^{-1}$ and 41.19 ± 9.62 µg $g^{-1}$ respectively (p = 0.02), whereas all other CPE contents present intermediate values around 35.03 ± 1.66 µg $g^{-1}$. The C-CPE/BPC (Fig. 2b) in the uppermost half centimetre varies significantly among the stations (one-way ANOVA, F = 76.82, p < 0.001). In particular, the algal fraction is significantly lower in the outer fjord station MC7 (4.09 ± 0.33%) compared to all other stations (p < 0.001). On the contrary, all the stations

inside the fjord do not differ significantly and have values between 13% and 17%.

**4.4 Foraminiferal assemblages of the 0-5 cm sediment layer (>150 µm fraction)**

**4.4.1 Abundances and diversity**

Considering the total foraminiferal faunas in the 0-5 cm sediment interval (Table 2), the highest absolute abundance is displayed at the inner fjord station MC2 (2249 ind. 50$cm^{-2}$) whereas it is reduced by about half at the other two inner fjord

stations (1104 and 1353 ind. 50$cm^{-2}$at MC1 and MC3 respectively). The absolute abundance increases in the deep basin stations (1861 and 1439 ind. 50$cm^{-2}$ at MC4 and MC5 respectively) and drastically declines at the sill station MC6 reaching the lowest abundance detected in the transect (940 ind.50$cm^{-2}$). At the outer fjord station MC7, the total absolute abundance is 1238 ind.50$cm^{-2}$.



Both the inner fjord stations MC1 and MC2 present the same number of species (27) (Table 2), and similar Shannon-Wiener

index (H' = 1.61 and 1.48) and equitability (J = 0.49 and 0.45). The third inner fjord station MC3 is characterised by the lowest diversity (19 species and H' = 0.92) and the lowest equitability (J = 0.31) whereas the deep basin stations MC4 and MC5 show relative high H' (2.25 and 2.35, respectively) and J values (0.62 and 0.70, respectively). The sill station MC6 shows similar H' and J values compared with the deep basin stations (2.18 and 0.65 respectively). The outer fjord station MC7 shows the highest number of species (44) and H' index (2.40).

In terms of species composition (Fig. 3a), the inner fjord stations are mainly dominated by two calcareous species: *Elphidium excavatum subsp. clavatum* contributing for 22, 47 and 75% of the total fauna and *Nonionellina labradorica* for 51, 31 and 13% at MC1, MC2 and MC3, respectively. It can be noted that *E. excavatum subsp. clavatum* is predominant at MC2 and MC3 in contrast with the prevalence of *N. labradorica* at MC1. *Elphidium bartletti* is a secondary species at station MC2 (8%) however it contributes for less than 2% at the two other inner fjord stations. The deep basin stations (MC4 and MC5) are

dominated by various agglutinated species that contributes differently to the total assemblages. The relative abundance of *Recurvoides turbinatus* varies from 12 to 18%, *Reophax fusiformis* from 13 to 10% and *Reophax scorpiurus* from 27 to 17%, at MC4 and MC5 respectively. Both *Ammotium cassis* and *Labrospira crassimargo* are less numerous at MC4 compared to MC5 (2 and 11% for *A. cassis* and 6 and 11% for *L. crassimargo* respectively). The calcareous *Nonionellina labradorica* is still abundant at MC4 (20%) but less at MC5 (8%). The sill station MC6 shows similarity with the deep basin stations because

of the presence of the agglutinated *R. turbinatus* (10%) and *R. fusiformis* (24%) but it differs by the presence of the agglutinated *Adercotryma glomeratum* (29%). The outer fjord station (MC7) can be distinguished from all other stations by the exclusive presence of the two calcareous species *Globobulimina auriculata* and *Melonis barleeanus* (9 and 12% respectively) and by the major contribution of the agglutinated species *Lagenammina difflugiformis* (14%). Nevertheless, some species which are abundant inside the fjord are also present at station MC7 (e.g. *N. labradorica* 25%, *R. fusiformis* 6% and *R. scorpiurus* 15%).

**4.4.2 Agglutinated vs calcareous foraminifera (0-5 cm, >150 μm)**

The comparison between the relative abundances of calcareous and agglutinated species, considering the total living faunas in the 0-5 cm sediment interval (Fig. 3b), shows the strong dominance of calcareous species (between 91 and 94%) in the inner fjord stations (MC1, MC2 and MC3). The opposite is observed in the two deep basin stations (MC4 and MC5) and in the sill station (MC6) where the relative abundances of agglutinated foraminifera vary from 65 to 77%. In the outer fjord station,

MC7, calcareous species have higher proportions (60%) although they are not as dominant as at the inner fjord stations.

**4.4.3 Vertical distribution**

The foraminiferal absolute density displays an overall decreasing trend from the surface sediment down to 5 cm depth at all stations, except for the outer fjord station MC7 showing a peak at the 3-4 cm sediment interval, mainly determined by the calcareous species *Nonionellina labradorica* (Fig. 4). At the inner fjord station MC1, the foraminiferal assemblage in the top

half centimetre is mainly constituted of three species: *Cassidulina reniforme* (~28%), *Elphidium excavatum subsp. clavatum*





(~36%) and *Triloculina oblonga* (~17%) (Fig. 4a). From 0.5 cm down to 4 cm depth, *N. labradorica* is the dominant species. At the two other inner fjord stations (MC2 and MC3), the topmost half centimetre is dominated by *E. excavatum subsp. clavatum* (~66% and ~89%, respectively). At the MC2 station, the abundance of this species decreases in the deeper layers whereas *N. labradorica*'s density increases. The MC3 foraminiferal density strongly decreases in the second half centimetre,

from ~1830 to ~270 ind.50 cm$^{-3}$ and the fauna is dominated by *E. excavatum subsp. clavatum* in both sediment layers (~89% in the 0.0-0.5 cm layer and ~57% in the 0.5-1.0 cm layer). The deeper layers (from 1.5 to 5 cm) show very low abundances (<100 ind. 50cm$^{-3}$), mainly represented by *N. labradorica* and *E. excavatum subsp. clavatum*. In the two deep basins (MC4 and MC5; Fig. 4b), three agglutinated species are dominant in the first half centimetre of sediment: *Reophax scorpiurus*, *Reophax fusiformis* and *Recurvoides turbinatus* (respectively ~29, 15 and 15% at MC4, and ~20, 10 and 26% at MC5). The

fourth agglutinated species *Labrospira crassimargo* shows a high density in the first half centimetre at MC5 (~16%) and is less abundant at MC4 (~5%). In the second half centimetre of the sediment at station MC4, a similar species composition to that in the upper half centimetre is present (*R. scorpiurus* 36%, *R. fusiformis* 23% and *R. turbinatus* 10%) but in much lower absolute abundances (~681 ind.50cm$^{-3}$). At this station, the infaunal species *N. labradorica* is present in high relative abundance (~57%) in the 1-2 cm sediment intervals. In the deepest sediment layers, *L. crassimargo*, *R. scorpiurus*. and *R.*

*turbinatus* are dominant. Similarly, the 0.5-1 cm sediment layer at station MC5 is dominated by the same agglutinated species as those in the topmost layer (*R. scorpiurus* ~16%, *R. fusiformis* ~15%, *R. turbinatus* ~11%). In addition, *Ammotium cassis* (~37%), *L. crassimargo* (~7%) and *Lagenammina difflugiformis* (~5%) are found. The dominance of *Nonionella digitata* and *N. labradorica* characterises the intermediate infaunal microhabitat from 1.5 cm to 3 cm depth. Here again, the deeper layers (3 - 5 cm) are dominated by the same agglutinated species as in the topmost layers. At the sill station MC6, the dominant

species present in the uppermost centimetre are *R. fusiformis* (~38%), *R. turbinatus* (~16%) and *Adercotryma glomeratum* (~16%) (Fig. 4b). This later species is also present in relatively high abundances in the deeper sediment layers, from ~29% (in the 1.0-1.5 cm layer) to ~51% (in the 4-5 cm layer), together with *N. labradorica* (~20% at 1 - 3 cm depth), and *Reophax* spp. ( ~14 to 30% at 1-5 cm depth). At the outer fjord station MC7, the 0.0-0.5 cm sediment interval shows a dominance of *R. scorpiurus* (~29%), *R. fusiformis* (~11%) and *L. difflugiformis* (~22%) (Fig. 4c). In the second half centimetre, these species

are accompanied by the calcareous species *Melonis barleeanus* (~18%). This species shows subsurface peaks (~58 and 49% in the 1.0-1.5 cm and 1.5-2.0 cm layers respectively). From 1 to 5 cm depth, we observe the increasing presence of *Globobulimina auriculata* (~14 to 34%).

### 4.4.4 Multivariate analysis

The CCA analysis based on the foraminiferal data (0-5 cm, > 150 µm, total absolute densities of 15 species, with a relative

abundance > 5%) and 15 measured environmental variables, is presented in Fig. 5. Axes 1 and 2 explain nearly 90% of the total variance. This multivariate analysis clearly divides the stations into three groups based on the differences determined by the foraminiferal assemblages and the environmental variables. Axis 1 strongly separates station MC7 from the rest of the stations. This difference is mainly determined by bottom water parameters (T, S, pH), OPD, percentage of silt and water depth.





The negative correlation between the outer fjord station MC7 and the CPE content and algal fraction contributes also to separate

it from all other stations. Axis 2 clearly divides the other six stations into two groups: the inner fjord group (composed by stations MC1, MC2 and MC3), and the deep basin and sill group (stations MC4 and MC5, and station MC6). This separation is mainly based on the organic matter composition of the sediment. The inner fjord group of stations is positively correlated with the percentage of proteins, whereas the deep basin and sill group is mainly correlated with the percentage of carbohydrates and the biopolymeric carbon content. This group of stations MC4-MC6 is also positively correlated with the CPE content and

the algal fraction of BPC. These three groups are characterised by different foraminiferal assemblages. Four calcareous species *Cassidulina reniforme*, *Elphidium excavatum subsp. clavatum*, *Nonionellina labradorica*, and *Elphidium bartletti* characterise the inner fjord, whereas three agglutinated species *Labrospira crassimargo*, *Adercotryma glomeratum* and *Recurvoides turbinatus*, and one calcareous species *Nonionella digitata* define the deep basin and sill group. Finally, the exclusive presence of the two species *Melonis barleeanus* and *Globobulimina auriculata* characterise the outer fjord station MC7.

**4.5 Comparison between the 63-150 µm and >150 µm size fractions (0-1 cm)**

**4.5.1 Abundances and diversity**

Foraminiferal abundances considering the entire >63 µm fraction (63-150 µm + >150 µm fractions) in the topmost centimetre of the sediment are maximal at MC2 and MC4 (4610 and 3936 ind. 50cm$^{-2}$ respectively) while all other stations present values between 2496 (MC7) and 2864 ind. 50cm$^{-2}$ (MC5) (Fig. 6a). The small fraction (63-150 µm) is dominant at all stations and

particularly at MC1 (~81%), MC6 (~83%) and MC7 (~77%). The lowest contribution of the 63-150 µm fraction is recorded at MC3 (~59%). When considering only the largest fraction (>150 µm) at the first centimetre, MC2 (1467 ind. 50cm$^{-2}$) and MC4 (1132 ind.50cm$^{-2}$) still shows the highest abundances followed by stations MC3 and MC5. For the rest, values vary between 422 and 567 ind. 50cm$^{-2}$.

Regarding diversity values, no significant differences are found between H' and J indices in the two size fractions between the

inner fjord station MC1 and the outer fjord station MC7 (Fig. 6b, c). Stations MC2, MC3 and MC6 present lower H' and J values for the >150 µm fraction whereas the opposite is observed for stations MC4 and MC5.

In terms of species composition (Fig. 7), diversity is higher in the 63-150 µm fraction at stations MC2 and MC3 because of the additional presence of *Stainforthia feylingi*, *Spiroplectammina biformis* and *Textularia torquata*. However, at the three inner fjord stations (MC1, MC2, MC3), *Elphidium excavatum subsp. clavatum* (juveniles) is still dominant (~16, 30 and 66%,

respectively). Similarly, *Cassidulina reniforme* (juveniles) is still highly dominant (~38%) at station MC1 in the 63-150 µm fraction. At stations MC4 and MC5, the lower diversity of the 63-150 µm fraction is due to the strong dominance of *S. biformis* (~75 and 65% at MC4 and MC5, respectively), that is nearly absent in the large size fraction. At station MC7, the small size fraction is characterised by the presence of juveniles of *Cassidulina teretis* (~11%) and *Melonis barleeanus* (~8%), species that are also present in the >150 µm fraction (~5% and ~8% respectively). These species are accompanied by *Globocassidulina*

*subglobosa* (~21% in the 63-150 µm) and *Alabaminella weddellensis* (~27%) that are only present in the small size fraction.



### 4.5.2 Agglutinated vs calcareous foraminifera

The percentage of agglutinated forms is systematically higher in the entire fraction >63 µm compared to the >150 µm at stations MC1 to MC6 (Fig. 8). This is explained by the presence of the small-sized agglutinated species *Spiroplectammina biformis*, and other minor agglutinated species (*Cuneata arctica*, *Textularia torquata*, *Cribrostomoides* sp.). Conversely, the outer fjord
station MC7 shows the opposite pattern mainly because of the presence in the small fraction of calcareous species that are absent in the >150 µm size fraction (i.e. *Cassidulina teretis*, *Globocassidulina subglobosa*, *Alabaminella weddellensis*).

### 4.5.3 Multivariate analyses

Coordinate 2 of the nMDS analysis separates the >150 µm fraction from the 63-150 µm and >63 µm fractions (Fig. 9a), and the cluster (Bray-Curtis similarity) analysis shows less than 50% of similarity between the two groups (Fig. 9b). However,
nMDS-Coordinate 1 groups all fractions into the three same stations' groups (Fig. 9a) previously determined by the >150 µm-CCA analysis (Fig. 5).

### 4.6 Visual characterisation of test dissolution

At all stations inside the fjord (from MC1 to MC6) most of the calcareous species display different degrees of dissolution including for small sized specimens. As visualised in the Fig.S2, the species *Elphidium excavatum subsp. clavatum, Elphidium*
*bartletti*, *Triloculina oblonga* and *Robertinoides sp.* show the most severe degree of dissolution, whereas *Nonionellina labradorica* seems to be less sensitive to dissolution (individuals classified at stage I or II of dissolution). Moreover, the highest degree of dissolution (stage IV) is observed only in the deep basin and the sill stations.

### 5 Discussion

### 5.1 Environmental characteristics of the study area

According to the topography of the fjord, distribution of the main water masses and physicochemical characteristics of the sediments, we can separate the fjord into three main areas: the inner fjord (i.e., stations MC1 to MC3), the central deep basins (i.e., stations MC4 and MC5) constrained by the sill (i.e. station MC6) and the outer fjord Storfjordrenna (i.e., station MC7) (Fig. 1b).

In July 2016, the inner fjord sea surface water temperatures and salinities (Fig. S3), are indicative of a mixture between Melt
Waters (MW) and Storfjorden surface water (SSW) as previously indicated by Skogseth et al. (2005b). On the contrary, the inner fjord bottom water parameters recorded are not homogeneous (Table 1). In fact, the stations MC1 and MC2, on the western side of the fjord, are characterised by salinity and temperature within the range of BSW as defined Skogseth et al. (2005b). The location of these stations in small topographic depressions on the shelf may explain the presence of these cold





and salty waters. The shallowest station MC3 (99 m depth), located on the eastern side of the inner fjord, seems influenced by
Modified Atlantic water (as defined Skogseth et al., 2005b).

The bottom water values of salinity and temperature measured in July 2016 in the deep basins, allow to identify the presence
of trapped residual BSW still long time after the season of sea ice formation  as previously hypothesised by Skogseth et al.
(2005b). The bottom water properties at the sill fall into the range of Arctic water (Skogseth et al., 2005b). In contrast with the
inner fjord and the deep basins, the outer fjord water column displays typical values of NAW from the surface to the bottom
(Skogseth et al., 2005b).

The summer melting of continental glaciers flowing in Storfjorden produces an important supply of terrigenous materials to
the head of the fjord (Winkelmann and Knies, 2005). This sedimentary dynamics results in relatively high sedimentation rates
of about $3.2 \pm 0.5$ mm yr$^{-1}$ recorded in the fjord (stations MC1 to MC6, Table S1), although lower than in other Svalbard fjords
(e.g. Kongsfjorden 5-10 mm.y$^{-1}$; Zaborska et al. 2006). Associated to this terrigenous flux, organic matter supply is high in the
internal fjord. In contrast, low sedimentation rate ($1.3 \pm 0.6$ mm yr$^{-1}$, Table S1) and low organic matter supply are recorded in
the outer fjord. This clearly indicates a lower influence of continental glacier's inputs in Storfjordrenna (station MC7)
compared to the internal fjord (stations MC1-MC6).

Regarding the organic supply, the high concentrations of organic matter at all our stations and particularly in the deep basins,
confirm the sedimentary organic-rich character of Storfjorden as previously reported in the literature (Winkelmann & Knies
2005; Mackensen et al. 2017). The higher CHO (%) associated to lower PRT (%) in the deep basins (station MC4 and MC5),
compared to the other stations, is indicative of the presence of older and more refractory organic matter (Pusceddu et al., 2000).
This could be either related to higher continental supplies of more refractory organic matter, heterotrophic nutrition, and/or the
presence of long-residence water masses, influenced by BSW and isolated by a strong chemocline during periods of sea ice
melting (Rysgaard et al., 2011). On the opposite, the higher contents of PRT (%) and CPE in the inner fjord (stations MC1-
MC3) and at the sill (MC6) could be the result of a recent (summer) phytoplankton bloom. Contrary to the inner fjord, the CPE
contents in the Storfjordrenna are much lower (Fig. 2b), indicating a lesser fresh algal input to the bottom, which is consistent
with the greater water depth (>300 m) at this outer fjord site.

The oxygen profiles and particularly the OPD (Fig. S1a) reflect the quantity of organic matter supplies. Indeed, the organic
carbon accumulation depends on its reactivity with available oxygen (Dauwe et al., 2001) and *vice-versa,* the oxygen
consumption is proportional to the organic matter mineralisation rate. Except for the outer fjord, all stations (MC1-MC6)
present shallow OPD values (<10 mm) in consistence with the high contents of available organic matter (i.e., BPC) (Fig. 2).
Organic matter aerobic respiration is also the reason for the rapid pH decrease in the first mm of the sediment column at these
stations (Fig. S1b). At the outer fjord, the slower pH decrease and the higher OPD (>15 mm) would be therefore attributed to
lower BCP contents at this station.



### 5.2 Distribution of foraminiferal species in response to environmental conditions

The foraminiferal distribution and the measured environmental parameters define three biozones: i) the inner fjord, ii) the deep basins and sill and iii) the outer fjord. The CCA analysis (Fig. 5) shows that the inner fjord faunas are positively correlated to PRT (%) and negatively correlated to CHO (%), meaning that they favourably respond to the availablility of fresh and labile organic matter. *Cassidulina reniforme* and *Elphidium excavatum* subsp. *clavatum* dominate the innermost fjord station MC1 where they occupy the superficial infaunal microhabitats. The high dominance of these species in the inner fjord is consistent with previous findings from other glacier-proximal inner Svalbard fjords (Hald and Korsun, 1997; Korsun and Hald, 2000) and Arctic domains (e.g., Iceland; Jennings et al., 2004). The species *C. reniforme* is known for its tolerance to high concentration of suspended particulate organic matter (Schäfer and Cole, 1986) and its preference for cold water environments (Jernas et al., 2018). The relationship between these two species in the glaciomarine environments is however not clear. Korsun and Hald (1998) hypothesised that *C. reniforme* becomes dominant when better conditions are present, such as a lower turbidity due to less glacier-driven sediment input and an increased phytoplankton production. In the two other inner fjord stations, MC2 and MC3, *C. reniforme* represents less than 1% of the living assemblage (>150 μm), and *E. excavatum subsp. clavatum* largely dominates the topmost half centimetre at both sites. The species *E. excavatum subsp. clavatum* is often described as able to adapt to harsh environments such as near tidewater-glacier fronts and riverine estuaries (e.g., Hald and Korsun 1997; Korsun and Hald, 1998; Forwick et al., 2010). According to these findings, faunas at stations MC2 and MC3 potentially reflect a more stressful environment than nearer the fjord head (station MC1). Additionally, the low diversity in the inner fjord mainly determined by the strong dominance of these two species, could be the result of a recent event of reproduction as confirmed by the high abundance of juveniles in the 63-150 μm fraction (>50 % at stations MC1 and MC3). This opportunistic behaviour may be a quick response to fresh organic matter input to the seafloor.

The intermediate microhabitats at the three inner fjord stations are dominated by *Nonionellina labradorica*. The ecology of this species is not very well constrained especially in glaciomarine environments. It is described as an intermediate-deep endobenthic species, found in high productivity area (Lloyd, 2006) because of its preference to feed on fresh phytodetritus and in particular on diatoms (Cedhagen, 1991 ). Korsun & Hald (2000) suggest that this species may start reproducing during spring in glaciomarine environments possibly following the diatom bloom starting in March under the sea ice (Rysgaard et al., 2011). In some studies, *N. labradorica* is also reported as an Atlantic Water indicator (Hald and Korsun, 1997; Lloyd, 2006). A few paleoceanographical studies draw parallels between its high abundances in the sediment and intensified Atlantic intermediate water circulation (e.g. Łącka & Zajączkowski, 2016; Rasmussen and Thomsen, 2015). In our study, this species occurs in the glacier-proximal areas (i.e., inner fjord), that were not influenced by Atlantic Water (AW) inflow during the sampling period in July 2016 (except at station MC3 which may be influenced by the MAW; see Fig. S3). Therefore, we rather interpret its presence as a response to meltwater discharge and consequent phytoplanktonic bloom. *Elphidium bartletti* occurs as an accessory species (especially at station MC2). Polyak et al. (2002) found this species in river-affected habitats of the southern Kara Sea explaining the higher frequency of E. *bartletti* in the off-shore part as dependent on the presence of coarser-





grained sediment or on several environmental variables (low salinity, high food availability and high sedimentation rate). We suppose that the presence of *E. bartletti* in the glacier-proximal area is determined by the availability of fresh food present in

July 2016 (high percentages of PRT), but it is probably suffering from competition with the more opportunistic species *C. reniforme* and *E. excavatum subsp. clavatum.*

The CCA analysis (Fig. 6) shows a negative correlation of the deep basin and sill faunas with the percentages of proteins and a positive correlation with the percentage of carbohydrates, meaning a targeted response of the fauna to old (refractory) organic matter. The assemblages in the two deep basin stations (MC4 and MC5) display a similar diversity which in both cases is

higher than in the inner fjord. Agglutinated species dominate (>71%) the topmost centimetre of the sediment, in particular the two species *Reophax scorpiurus, Reophax fusiformis* and *Recurvoides turbinatus*. These species are often found in the distal part of other Svalbard fjords (e.g., Hald and Korsun, 1997; Murray and Alve, 2011; Jernas et al., 2018) and are considered to tolerate low food quality, high sedimentation rates, and a wide range of salinities, temperatures and organic matter fluxes (Hald and Korsun, 1997; Murray, 2006; Jernas et al., 2018). Other accessory species such as *Labrospira crassimargo* and *Ammotium*

*cassis* are present in relatively high abundances at station MC5. This group of agglutinated species found in the deep basins, includes some of the most adaptable species to largely different ecological conditions (Murray and Alve, 2011), and is described as widely distributed in areas covered with seasonal sea ice in the Arctic Ocean (e.g. Wollenburg and Kuhnt, 2000; Wollenburg and Mackensen, 1998). This capacity of adaptation may explain the abundance of all these agglutinated species in the deep basins where residual BSW (with high S, low T and pH) and high concentrations of refractory carbohydrates are

recorded in the top sediment. At station MC4, *N. labradorica* lives also in subsurface microhabitat (from 1 down to 3 cm depth) probably profiting from the phytodetritus supply testified by high CPE contents. The assemblage composition at the sill (station MC6) is close to those found in the deep basins, except for the dominant presence of *Adercotryma glomeratum* (at each sediment layer at this site). This species is considered as an opportunistic taxon (e.g. Gooday and Rathburn, 1999; Heinz et al. 2002), and it has been reported as positively related to increasing distance from glaciers and from the fjord head (Hald

and Korsun, 1997). This feature suggests a positive relationship with an increase in salinities and temperatures and consequently with Transformed Atlantic Water (TAW) (Hald & Korsun, 1997; Jernas et al., 2018). The dominance of *A. glomeratum* at the sill and the concomitant similarities with the deep basin assemblages could suggest a seasonal alternation of water mass influences between summer incursion of Atlantic Water and winter overflow of BSW during brine production. Moreover, the presence of very badly preserved calcareous foraminifera tests and the dominance of agglutinated species in the

deep basins plus the sill, strongly suggest that acidic BSW is a primary control in bottom water ecology in this sector of the fjord.

The outer fjord biozone (station MC7) is characterised by a deeper water column (> 300 m), and higher bottom water salinity and temperature compared to the rest of the studied sites. The clear predominance of the NAW at this site is indicated by the presence of typical Atlantic species such as *Melonis barleeanus* and *Globobulimina auriculata*. In high latitudes, *M.*

*barleeanus* is described as an arctic-boreal infaunal taxon, whose presence suggests the influence of relatively warm AW (Caralp, 1989; Polyak et al., 2002; Jennings et al., 2004; Knudsen et al., 2012; Melis et al., 2018). In the Atlantic Ocean, this





species is described as intermediate infaunal, with an opportunistic behaviour in response to good quality organic matter (e.g., Nardelli et al., 2010). *Globobulimina auriculata* is also an infaunal species presumed to be related with increasing bottom water salinity (Williamson et al., 1984; Jernas et al., 2018).

### 5.3 Agglutinated vs Calcareous taxa: the premise of a paleo-proxy of brine formation

In both size fractions (63-150 µm and >150 µm), low agglutinated/calcareous (A/C) ratios characterise the inner and outer fjord in contrast with the high values observed at the deep basin and sill stations (Fig. 3b; Fig. 8). The exclusive presence of some agglutinated species in the smaller fraction (e.g. *Cuneata arctica*, *Spiroplectammina biformis* and *Textularia torquata*) results in relatively higher A/C ratios for the > 63 µm compared to the > 150 µm. It is worth mentioning that, in the Storfjordrenna (station MC7), the A/C ratio in the >63µm is similar to that found in the >150 µm fraction (Fig. 8), despite the presence of several small calcareous species (e.g. *Stainforthia feylingi*, *Cassidulina teretis*, *Alabaminella weddellensis*, *Globocassidulina subglobosa*) that are not present in the larger fraction.

Several hypotheses, which are eventually not exclusive, arise to explain the dominance of agglutinated species at the deep basin and sill stations:

(i) Jernas et al. (2018) suggest that agglutinated species may be more tolerant or less sensitive to the lower quality and/or quantity of food than the calcareous fauna. Following this idea, the dominance of agglutinated species in the deep basins is coherent with the more refractory organic matter (higher CHO %) measured in July 2016 (Fig. 2). However, the lower percentage of CHO observed at the sill station MC6 seems to contradict this hypothesis.

(ii) Low relative abundances of calcareous taxa found in the sediment in the deep basins could be attributed either to hampered growth, limited reproduction, and/or to test dissolution, which in the latter case leads automatically to an overestimation of the agglutinated populations. Indeed, we observe a severe degree of dissolution on living specimens of *Elphidium excavatum* subsp. *clavatum*, *Elphidium bartletti*, *Triloculina oblonga* and *Robertinoides* spp. found at the deep basin stations (Fig. S2). The most obvious explanation for the observed severe dissolution is the effect of brine waters, that persist all year round at stations MC4 and MC5 and may impact station MC6 through episodes of overflow from autumn to spring. In the inner fjord stations, these species present also some dissolution but less severe than at the deep-basin and sill stations (Fig. S2). This may be related to the high seasonal input of meltwater as a factor affecting the preservation of carbonate (Schroder-Adams et al., 1990). Previous studies in the Barents Sea related the prevalence of agglutinated faunas to carbonate dissolution in areas influenced by cold waters (Hald and Steinsund, 1992; Steinsund and Hald, 1994). The same conclusion was drawn in the fjord-shelf areas off eastern Greenland where agglutinated foraminifera are exclusively present beneath Polar Water and mixed assemblages beneath Atlantic Water (Jennings and Helgadottir, 1994). In some paleoceanographical studies from the Arctic, the high proportion of agglutinated taxa in sediment cores was considered to be dependent on bottom-water hydrographical condition (Seidenkrantz et al., 2007) and specifically on corrosive brine production (Rasmussen and Thomsen, 2015). Rasmussen and Thomsen (2015) inferred that the agglutinated species *Reophax scorpiurus* and *Adercotryma glomeratum* may





tolerate the $CO_2$-rich conditions characterising the brine environment because they are the most abundant species found in the

historical record from the deep basins of the fjord.

(iii) As an alternative, or in parallel, calcareous test dissolution may have resulted from decaying organic matter. Indeed, test dissolution of *E. excavatum* subsp. *clavatum* species was previously observed in the Adventfjorden (west Svalbard) and was attributed to low pH in the pore waters of upper sediments due to organic matter decay (Majewski and Zajaczkowski, 2007), whereas in the Barents-Kara shelf this process was associated to sinking of brines (Hald and Steinsund, 1992; Steinsund and

Hald, 1994).

To sum up, stressful ambient conditions (i.e. corrosive waters, low quality food) were measured in the deep basin and at the sill stations in summer 2016, and the associated foraminiferal assemblages display high A/C ratios. This result strongly supports the high potential of using the A/C ratio as a proxy for brine persistence or/and overflow.

### 5.4 Insights from the small size fraction

The additional observation of the small size fraction (63-150 µm) results in the definition of the same three biozones and similar A/C ratios as for the larger fraction (Figs. 8; 9a). The cluster analysis (Fig. 9b) further shows that the consideration of the 63-150 µm fraction increases the percentage of similarity among stations belonging to the same biozone and it increases the dissimilarity between the stations inside the fjord (from MC1 to MC6) and the outer fjord (MC7). Nonetheless, the study of the 63-150 µm fraction provides new insights into the ecology of Storfjorden small-sized species that are not found in the

larger fraction.

At the inner fjord stations MC2 and MC3, the exclusive presence of the calcareous species *Stainforthia feylingi*, and the three agglutinated *Cuneata arctica*, *Spiroplectammina biformis* and *Textularia torquata* in the 63-150 µm fraction increases the overall diversity compared with the large dominance of *Elphidium excavatum subsp. clavatum* in the larger fraction. These four species with individuals of small size, are typical of Arctic and cold boreal environments showing an opportunistic

behaviour in response to a wide range of environmental conditions (e.g., Schäfer and Cole, 1986; Hald and Korsun, 1997; Korsun & Hald 1998, 2000; Lloyd et al., 2007; Leduc et al., 2002; Pawlowska et al., 2016; Jernas et al., 2018). The presence of numerous juveniles (63-150 µm) of *E. excavatum* subsp. *clavatum* and *Cassidulina reniforme* in the inner fjord suggests also an opportunistic response of the fauna, possibly related to a recent phytoplanktonic bloom and associated fresh organic matter input in the benthic system, as suggested by the high percentages of PRT measured in the sediment (Fig. 2).

On the contrary, in the deep basins, the diversity decreases when the small fraction is considered, due to the strong dominance of the agglutinated species *S. biformis*. In the literature, this opportunistic species is characteristic of glaciomarine habitats, and is usually found in the outer part of fjords as indicative of the presence of cold arctic waters (Hald and Korsun, 1997; Korsun and Hald, 1998, 2000; Schäfer and Cole, 1986). The high numbers of small individuals found in the deep basins coupled with the high percentages of CHO may suggest an eventual positive response of *S. biformis* to refractory organic

matter. The relatively high abundances of *T. torquata* at the sill station MC6 suggests high salinity fluctuations (Wollenburg and Kuhnt, 2000), which could be consistent with occasional/seasonal overflow of BSW at this site.





In the Storfjordrenna, *Cassidulina teretis*, *Globocassidulina subglobosa*, and *Alabaminella weddellensis*, are exclusively present in the 63-150 µm fraction. These three species, usually associated to AW (Wollenburg & Mackensen, 1998), give evidence of the dominant influence of this water mass in the outer fjord area. However, the inclusion of these species in the
estimation of diversity does not change substantially the results obtained from the >150 µm fraction.

Taking into consideration the comparison between the 63-150 µm and >150 µm data, the additional information from the small fauna is limited to a bit more precise estimation of biodiversity and the confirmation of ecological speculations based on the large fauna (e.g., recent blooming events at inner stations). Therefore, and regarding the high time-consuming character inherent to the investigation of the 63-150 µm fraction, we propose that the small fraction could be neglected in comparable
future studies in Storfjorden unless in the aim of answering some very specific questions.

## 6 Conclusion

Living benthic (rose Bengal stained) foraminiferal faunas from Storfjorden "sea ice factory" were studied in order to determine the response of foraminiferal communities to the major driving factors controlling the sea bottom ecology in this area (e.g. bottom water properties, sediment characteristics, organic matter quantity and composition, sediment oxygen profiles, pH).
The benthic ambient conditions were further connected to Brine-enriched Shelf Waters (BSW) and indirectly to sea ice production.

The influence of the BSW persistence on benthic foraminiferal assemblages was identified on the base of characteristic faunas inhabiting the two deep basins and the sill of the fjord. At these sites, BSW are respectively trapped for a long part of the year or overflow during the maximum production period. The assemblages at these stations are dominated by agglutinated taxa and
the presence of heavily dissolved calcareous tests supports the hypothesis that one of the main responsible for this result is the corrosive character of BSW. Also the chemocline related to BSW presence at the bottom could limit the fresh organic matter flux to the seabed and indirectly influence the assemblages.

These stations have very different faunas if compared to the "inner fjord" where the biozone is characterised by calcareous faunas and in particular by typical glacier proximal species, able to tolerate turbidity caused by glacier-driven sediment input,
and responding to fresh and labile organic matter inputs (summer phytoplankton bloom) to the seafloor after the melting of sea-ice.

Outside the fjord, the biozone shows species composition partly in common with the "inner fjord" and the "deep basins and sill", suggesting the influence of outflow from the fjord to the Storfjordrenna. However, this area is also characterised by the exclusive presence of typical North Atlantic species confirming the strong influence of the NAW in the area.
In the light of these results, the low A/C ratios characterising the "inner fjord" in opposition to the high A/C ratios found in the "deep basins and sill" let suggest the potential use of A/C ratio as a proxy for brine persistence and overflow causing stressful conditions (i.e. acidic waters and low food quality). We thus suggest that the A/C proxy can be applied on historical

sedimentary records from Storfjorden in order to reconstruct past changes in BSW intensity and, by extent, in sea ice production.

**Supplement**

Table S1, Table S2, Fig. S1, Fig. S2 and Fig. S3 as referred in the manuscript can be found in Supplementary Material. Scanning electron micrographs (plates) of the most relevant species are shown in Fig. S4 and Fig. S5 in the Supplementary Material.

**Data availability**

Raw data are available in Supplementary Material (Table S3, S4).

**Author contributions**

EF, MM, MPN and HH wrote the manuscript which was commented by all co-authors. EM was the cruise leader and field work was performed by HH and BL. EF, MPN, MM, HH, AJ, BL and DM collected the data and EF, MPN, MM, HH, AJ, BL and AP analysed and interpreted the data.

**Competing interests**

The authors declare that they have no conflict of interest.

**Acknowledgements**

We are grateful to the captain and crew of R/V *L'Atalante*, chartered by IFREMER (French Research Institute for Exploitation of the Sea), Frédéric Vivier (co-chief of the cruise), as well as all participants who have contributed to the success of to the STeP cruise. We particularly thank Bruno Bombled for his technical assistance during the cruise. Original SEM micrographs of foraminiferal species (Figure S2, S4, S5) were realized by Romain Mallet at SCIAM (Université d'Angers). We fully acknowledge the efficient technical help provided by Sophie Quinchard and Raphäel Hubert-Huard. The research was funded by the projects ABBA (Observatoire des Sciences de l'Univers de Nantes Atlantique) and Bi-SMART (University of Angers). This research is part of the PhD thesis of EF, which is co-funded by French National Program MOPGA (Make Our Planet Great Again) and the University of Angers.



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

| Sampling date | Station | Latitude (N) | Longitude (E) | Depth (m) | Temperature (°C) | Salinity | Density (kg m$^{-3}$) | pH$_T$ | O$_2$ (µmol.L$^{-1}$) |
|---|---|---|---|---|---|---|---|---|---|
| 13/07/2016 | MC1 | 78°15.0 | 19°30.0 | 108.0 | -1.74 | 34.89 | 1028.59 | 8.00 | 341 |
| 14/07/2016 | MC2 | 77°50.0 | 18°48.0 | 117.0 | -1.59 | 34.79 | 1028.52 | 7.95 | 317 |
| 14/07/2016 | MC3 | 77°58.6 | 20°14.6 | 99.0 | 1.10 | 34.74 | 1028.29 | 8.12 | 350 |
| 15/07/2016 | MC4 | 77°29.2 | 19°10.6 | 191.5 | -1.78 | 34.92 | 1029.01 | 7.92 | 319 |
| 17/07/2016 | MC5 | 77°13.2 | 19°17.9 | 171.0 | -1.78 | 34.93 | 1028.91 | 7.91 | 317 |
| 18/07/2016 | MC6 | 76°53.9 | 19°30.3 | 157.0 | -1.13 | 34.80 | 1028.72 | 7.97 | 317 |
| 19/07/2016 | MC7 | 76°00.9 | 17°03.4 | 321.0 | 3.53 | 35.05 | 1029.33 | 8.04 | 305 |


**Table 1: Geographic coordinates, depths of the seven studied stations and bottom water parameters (temperature, and salinity measured in situ by the CTD, O$_2$ = dissolved oxygen and pH$_T$ measured from Niskin bottles).**




| Stations | MC1 | MC2 | MC3 | MC4 | MC5 | MC6 | MC7 |
|---|---|---|---|---|---|---|---|
| Abundance (ind. 50cm$^{-2}$) | 1104 | 2249 | 1353 | 1861 | 1439 | 940 | 1238 |
| Species Richness | 27 | 27 | 19 | 37 | 29 | 29 | 44 |
| Shannon-Wiener (H') | 1.61 | 1.48 | 0.92 | 2.25 | 2.35 | 2.18 | 2.40 |
| Equitability (J) | 0.49 | 0.45 | 0.31 | 0.62 | 0.70 | 0.65 | 0.64 |


**Table 2:** Foraminiferal total abundances (in number of individuals per 50 cm$^2$) and diversity indexes, considering the total living faunas (>150 µm size fraction) in the 0 to 5 cm core top sediment.

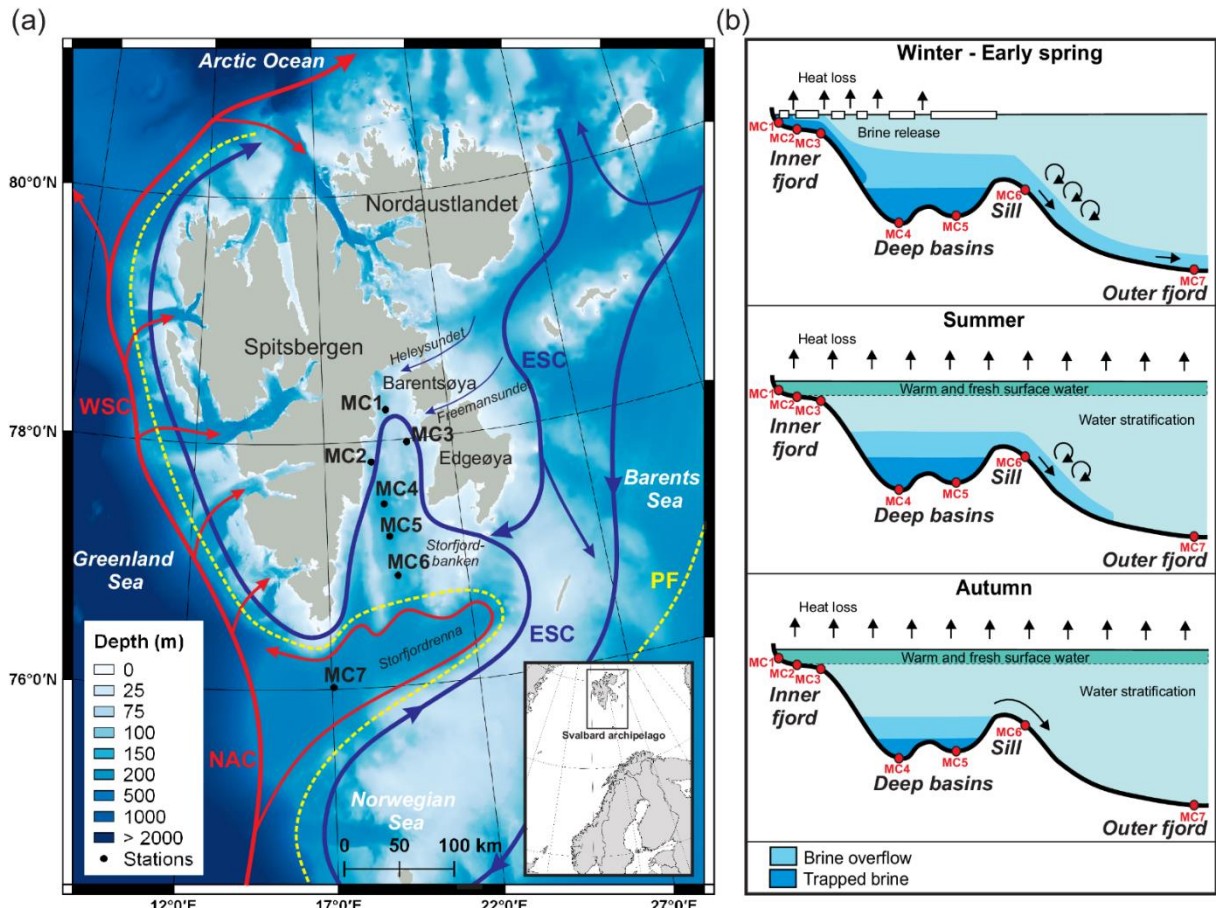

**Figure 1:** a) Bathymetric map showing the main current circulation around the Svalbard archipelago (modified from Skogseth et al. 2005b and Misund et al. 2016) and location of the sampling stations. The red lines represent the warm North Atlantic waters carried by the Norwegian Atlantic Current (NAC) and West Spitsbergen Current (WSC). The blue lines represent the cold Arctic waters carried by the East Spitsbergen Current (ESC). Dotted yellow line represents the Polar front (PF). Bathymetry obtained from EMODnet (http://portal.emodnet-bathymetry.eu) and map elaborated with QGIS (made with Natural Earth). b) Longitudinal
bathymetric profile sketches showing seasonal formation and flow of brines in the inner and outer Storfjorden (modified from Skogseth et al. 2005a and Rasmussen & Thomsen 2015) and indicative location of the sampling stations (red dots).



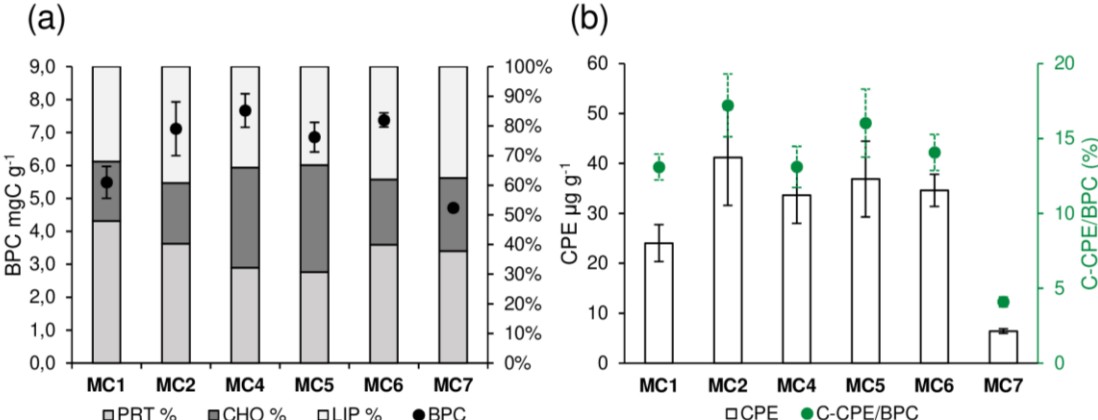

**Figure 2:** For each sampling station (data are not available at station MC3): a) Content of biopolymeric carbon (BPC, black dots) and percentage of protein, carbohydrate and lipid (cumulative bars). b) Content of chloroplastic pigment equivalents (CPE, white bars) and algal fraction of BPC (C-CPE/BPC, green dots).

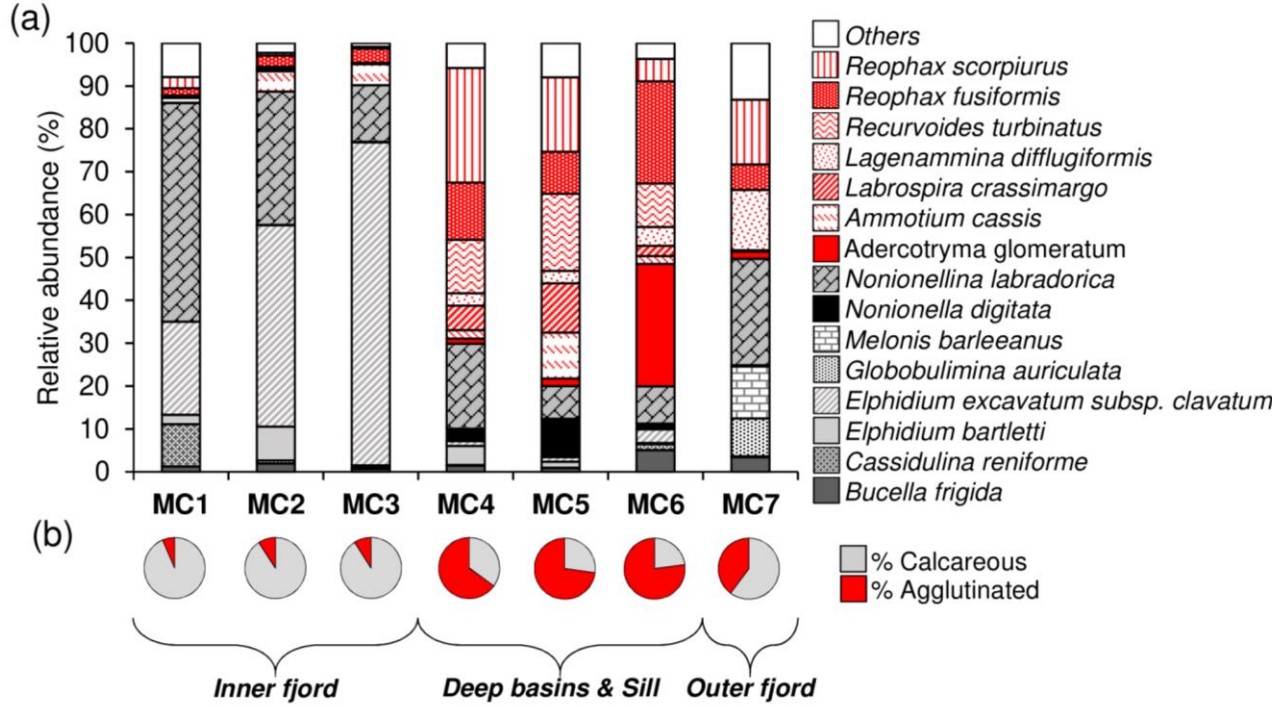

**Figure 3:** a) Species relative abundances of the total living faunas (>150 μm fraction) in the 0 to 5 cm core top sediment at each station and b) agglutinated species (in red) vs calcareous species (in grey) ratio.




**Figure 4:** Foraminiferal vertical distribution from 0 down to 5 cm sediment depth (>150 μm fraction) for a) the inner fjord (stations MC1 to MC3) b) deep basins (stations MC4 and MC5) and sill (station MC6) and c) outer fjord (station MC7). Calcareous species are shown using different grey textures, whereas agglutinated species using different red textures. The dashed black line represents the average oxygen penetration depth (OPD) at each station.





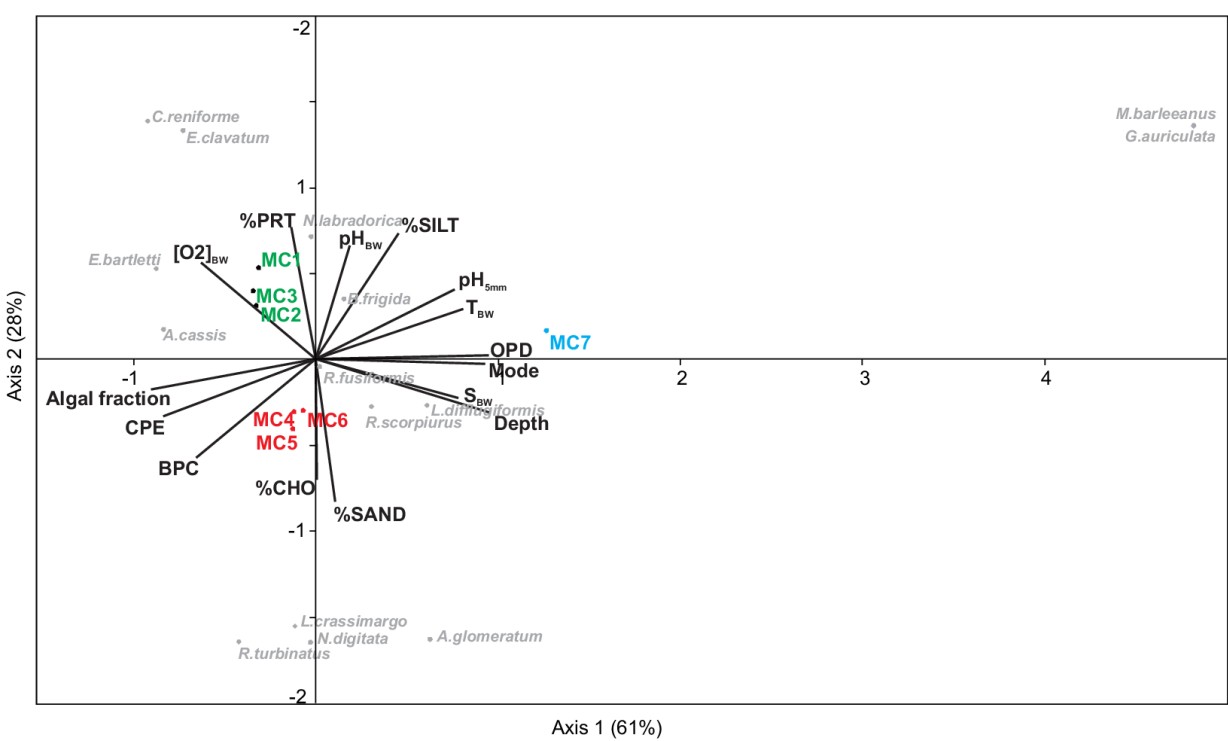


**Figure 5:** Canonical Correspondence Analysis based on real abundances (ind. 50 cm$^{-2}$) of the living faunas in the 0-5 cm sediment layer (> 150 μm size fraction) considering the major species (> 5%) vs environmental variables described in tables 1, S2, S3.

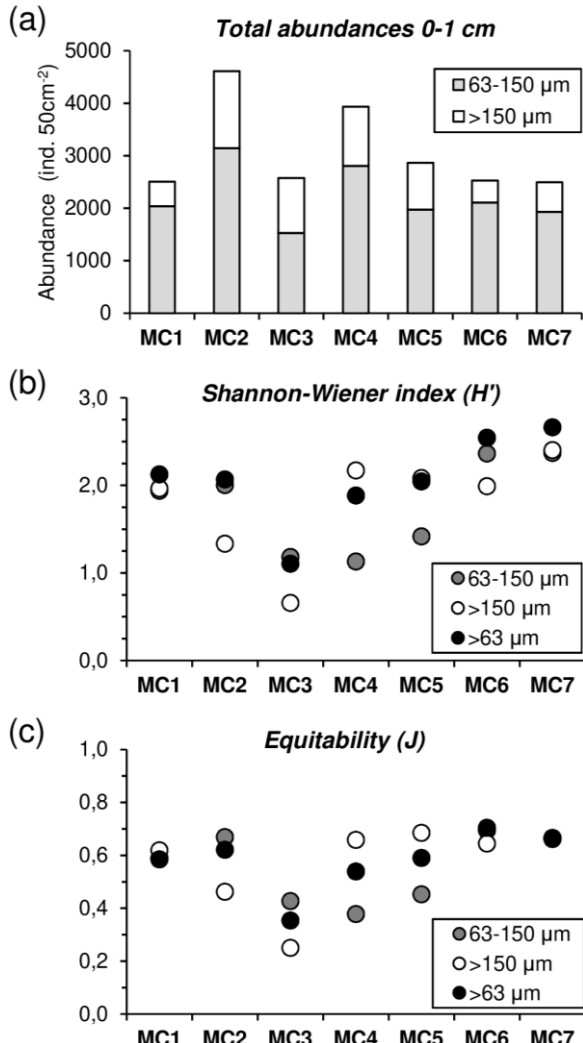

**Figure 6:** a) Foraminiferal cumulative abundances (ind. 50 cm$^{-2}$) for two size fractions (63-150 µm, grey, and >150 µm, white) of the
0-1 cm sediment layer. b) Shannon-Wiener (H') and c) Equitability (J) indexes comparison among the 63-150 µm, the >150 µm and
the >63 µm (black).



**Figure 7:** Total abundances (ind. 50 cm⁻²) and correspondent relative abundances (%) of the dominant species (>5% in at least one station) of the 0-1 cm sediment layer for the 63-150 µm fraction (a, c) and the >150 µm fraction (b, d). The calcareous species are shown using different grey textures, whereas agglutinated species using different red textures.



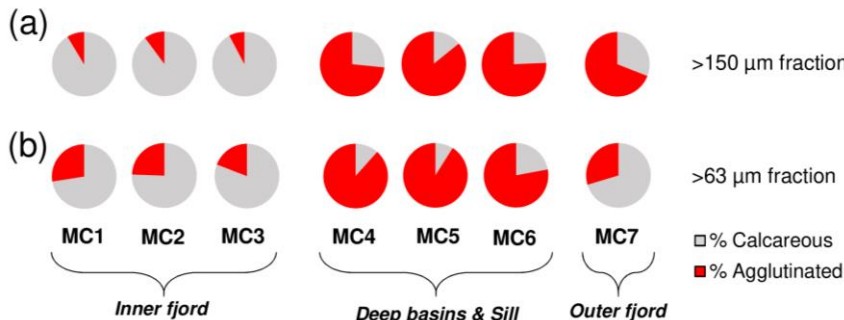

**Figure 8:** Relative abundances of calcareous foraminifera (in grey) and agglutinated foraminifera (in red) considering the > 150 µm size fraction (a) and the > 63µm size fraction (b).


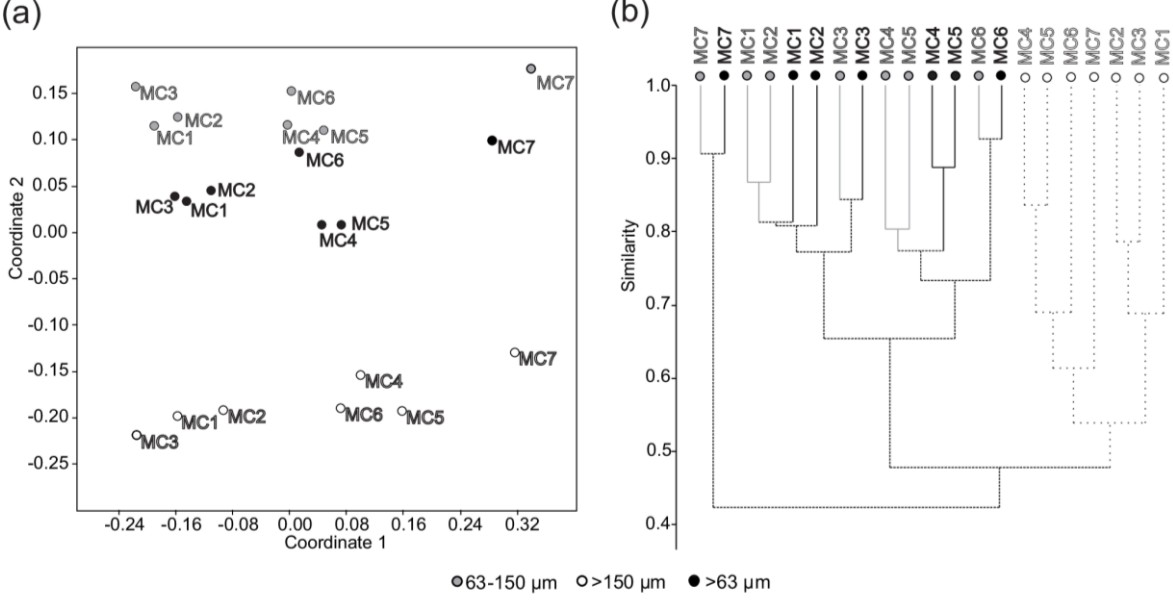

**Figure 9:** a) Non-metric multidimensional scaling analysis and b) cluster analysis (Bray-Curtis similarity measure) considering the densities (ind. 50 cm⁻²) of the major foraminiferal species (relative abundance > 5% in at least one station in one size fraction) for the 63-150 µm fraction, the >150 µm and the total fraction >63 µm.
