# Peer review of "Benthic foraminifera as tracers of brine production in Storfjorden "sea ice factory""

_Biogeosciences, 2019_

## Referee Comment (RC1) · Shungo Kawagata (Referee) · 5 Dec 2019

bg-2019-405    Submitted on 04 Oct 2019

Benthic foraminifera as tracers of brine production in Storfjorden "sea ice factory"

Eleonora Fossile, Maria Pia Nardelli, Arbia Jouini, Bruno Lansard, Antonio Pusceddu, Davide Moccia, Elisabeth Michel, Olivier Péron, Hélène Howa, and Meryem Mojtahid

The ms submitted for publication in Biogeosciences Discussions describes the distributions and environmental relationship of benthic foraminiferal fauna for 7 stations from the N–S transect in Storfjorden.  I consider this to be an interesting manuscript that represents a comprehensive benthic foraminiferal data and geochemical data. This manuscript will be of publishable quality once some revisions are made. The authors need to address each of the following points:

1.  The paper is fairly well-written, but too long. The presentation of the results and discussions are very tedious and can be shortened considerably.
2.  In Figure 1 (b), the longitudinal bathymetric profile should be redrawn to reflect the actual water depth. In particular, site MC3 is located at a deeper depth than sites MC1 and MC2 despite the shallowest water depth.
3.  The agglutinated species composition (>150 microns) at site MC7 is similar to those of sites MC4 and MC5, but different from site MC6. The authors interpreted that the faunal similarity among sites might be caused by overflowing of the corrosive bottom water from the deep basin passing through the sills, but no explanations for the cause of faunal differences between site MC6 and others. (1) Is corrosive bottom water likely to overflow the basin without passing through site MC6, or flow out of the fjord through another path (e.g., channel)? (2) Another question arises about the possibility that basin species once carried outside the fjord have settled at the MC7 site. If so, a scenario overflowing from corrosive bottom water basins is no longer needed.
4.  The authors conclude that Agglutinated/Calcareous (A/C) proxies are possibly useful for changes in past fjord BSW intensity and sea ice production. However, the past A/C in sediments do not always reflect the marine environment at that time because agglutinated tests are more fragile than calcareous ones in general and are less likely to be preserved as fossils. How do the authors think about this?
5.  Other comments

(1) In text, references and captions "and" and "&" are mixed.

(2) In text, figures, Tables and captions "subsp." is not required for "Elphidium excavatum subsp. clavatum".

(3) line 88: Publication year of "Haarpaintner et al., 2001" should be 2001a, 2001b or 2001c.

(4) line 89: "Polyakov et al., 2012" is missing in References.

(5) line 108: "Fer, 2004" is missing in References.

(6) lines 109, 120: Publication year of "Skogseth et al., 2005" should be 2005a or 2005b.

(7) line 204: "Pielou Index (1975)" needs author name(s) in the bracket.

(8) line 434: "Rysgaard et al., 2011" is missing in References.

(9) lines 536, 841: "Schroder-Adams" should be "Schröder-Adams".

(10) line 540: "Jennings et Helgadottir, 1994 " should be "Jennings and Helgadottir, 1994 ".

(11) line 662: Fer et al. (2004) is missing in the text.

(12) line 683: Haarpaintner  et al. (2001c) is missing in the text.

(13) line 704: Hunt and Corliss. (1993) is missing in the text.

(14) line 750: Swap "Lloyd et al. (2007)" and "Lloyd (2006)".

(15) lines 650, 660: Separate author's name with "and".

(16) Figure 4: In legend "Adercotryma glomerata" should be "Adercotryma glomeratum", "Cribrostomoides crassimargo" should be "Labrospira crassimargo".

(17) Figure S1: What does the difference in the color of the profile line at each sampling point indicate?

(18) Figure S4: For the taxonomy, SEM image of *Globocassidulina subglobosa* seems to be of *Cassidulina reniforme*.

NAME: Shungo Kawagata

AFFILIATION:  Yokohama National University, Japan

---

## Referee Comment (RC2) · Anonymous Referee #1 · 14 Jan 2020

The manuscript entitled 'Benthic foraminifera as tracers of brine production in Storfjorden "sea ice factory" by Eleanor Fossile and others definitely fits within the mission on Biogeosciences.

The hypothesis tested with the data presented is that the ratio of agglutinated to calcareous benthic foraminifera in Storfjorden, Svalbard archipelago, is largely controlled by brine formation and therefore can be used as a proxy for brine rejection processes and brine overflows in the paleo record. Previous research in the area has established that brine formation and overflow out of Storfjorden happen today and it has been inferred to have occurred in the past. The authors need to show that brine formation causes carbonate dissolution in modern samples and to rule out other processes such as high TOC causing low pH in the porewaters or show how other processes combine

with the brine formation and overflow to cause dissolution.

I can see that it is important to have a proxy for the brine formation in this area and that it relates to the coastal polynya and the sea-ice factory, but please state clearly why it would be important for paleo studies to know if there were brines forming or not in the past. Why is your study significant? What does brine formation tell us about the sea ice conditions or climate/environment in the larger Arctic?

You use a carefully developed dataset of living (biologically stained) foraminifera and environmental parameters such as various food types, hydrographic parameters and grain size from modern seabed samples to explore the ecology of the modern fauna. I find the paper to be carefully and clearly written in general. The development of the biozones and their association with various qualities of food source and hydrography is well done. However, I am not 100% convinced about the role of brine as the main driver of dissolution, but I think you could hone your arguments. I have some questions with that in mind. 1. Is there any chance of dissolution of calcareous faunas during your laboratory methods? Ethanol has a pH of 7.3. Did your samples sit in unbuffered water? You do not mention anything about buffering. 2. Are there dead (unstained) calcareous forams in your samples? It seems like an important missing bit of information. Are there large variations in calcareous forams in the fossil record? Is that how people have inferred that there were brines in the past? 3. The living fauna at the time of sample collection may not represent only this year or only one season, or it may exclude forams that bloomed earlier in the year. Give some insight into what the living fauna represents in terms of time. It does seem strange that living forams are badly dissolved! Are they really living at the time of collection or are they recently dead and already dissolving? 4. How do you determine what degree or type of staining points to a 'living' foram at the time of collection. 5. I suggest you add the word 'living' as a modifier of calcareous, agglutinated etc more often because you are only presenting living assemblages and that really needs to be made clear. For example 4.5.1 Abundances and diversity of living forams. 6. Can you provide a concise summary of why brine is

corrosive to CaCO3, along with explanation about how other factors (high CO2, cold Arctic water and high TOC for example) interplay or potentially play their own role in the dissolution?

Questions about the environmental setting:

Storfjorden is called a fjord and you mention often about glacial meltwater and its influence on the headwaters of the fjord. But glaciers are not shown on your maps and I don't think they are described in your paper. That description is needed because you call on glacial meltwater and sediment delivery as an important part of the environmental gradient. On line 421 and 425 you use the term continental glacier, but I think you mean plateau ice cap or mountain glaciers? And Storfjorden really looks like a sound as it forms a connection between the Barents and Greenland seas via Heleysundet and Freemansundet. What role do these connections play in the fjord hydrography? How important is the ESC waters that come into the head of the fjord for the formation of brines? Can the differences you see in MC3s be related to its proximity to Freemansundet? Your map figure is so small that I could not easily read the labels.

On Figure 1b, add the Atlantic Water..you can use a special arrow or something. Also add the Arctic water. Can you add the polynya to Figure 1a?

On page 4 you discuss the organic matter composition of the sediments and the potential of a terrestrial component. What is the bedrock geology of this area? Can some refractory carbon be from bedrock erosion and deposition in the fjord?

Line 125. How does Storfjorden introduce brine to the Arctic Ocean when it drains to the Greenland Sea?

Line 545. Explain what it is about brines that make them corrosive to carbonate. You have said they have high CO2 content, which is also what the Arctic Surface waters have that enter the fjord. I think you are getting at several factors that converge to make acidic pore waters in the fjord basins one of which is brines. Clarify and organize this

argument.

In your conclusions you also mash together the brine and other factors that can cause dissolution together (Lines 595 to 597) but the takeaway is that the dissolution is because of the brine formation. Can you clarify this and maybe state that brines are associated with some other conditions that converge to cause dissolution?

Minor comments: 45 in the meantime 105 clarify this sentence. If there is a persistent polynya then why is there extended winter sea ice cover as well? 138 (10 cm diameter) 150 microelectrode 162 'replicate analyses' Not replicated? I find this sentence unclear. How do you know which sample is most representative? 166 Pb dating was... 184 30 $\mu$g C $\mu$g phytopigment -1. Is this the correct way to state this? It is awkward 247 describe the silt % in various samples and its range in percent. Say that $20\mu$m is medium silt and 10 $\mu$m is fine silt. 249 and declines to 6.8% at MC7 258 not lower that (n)...have to say 'less than or equal to'... 297 Elphidium clavatum is considered to be a separate species now. See Darling et al., 2016 in Marine Micropaleo v. 129, p 1-23. P. 10 suggest you add the word living to modify foraminifera in this section. You need not do this every time, but use this modifier in the top of each section and especially in the heading so that it is clear that your total assemblage is limited to living fauna. 354 change 'distinguishes also to separate' to distinguishes 477 italicize E. 478 Melonis has been associated with degraded OM (Caralp, 1989) 479 G. auriculate is often associated with buried OM 533 I don't know if it is true that the most obvious explanation for the severe dissolution is the brine. You really need to build this argument. This lack of building an argument about the affect of brine and the other factors associated with Arctic water and TOC weakens the paper. 540 change et to and

Figures:

The stacked histograms showing species at sites are really hard to read. A major problem is that the key is so small that a person cannot see the pattern. I like the idea, but it may be better to make histograms of species in each site and stack them

one above the other. Or choose fewer species. For example you could use only the species found to be statistically significant.

---

## Author Comment (AC1) · 27 Jan 2020

**Referee #2 Shungo Kawagata**

**Major comments**

**1_R#2:** *The paper is fairly well-written, but too long. The presentation of the results and discussions are very tedious and can be shortened considerably.*

**Answer:** We thank the referee for the positive comments. We will shorten the manuscript as much as possible considering also the comments of the reviewer #1, especially the discussion about the ecological preferences of the benthic species.

**2_R#2:** *In Figure 1 (b), the longitudinal bathymetric profile should be redrawn to reflect the actual water depth. In particular, site MC3 is located at a deeper depth than sites MC1 and MC2 despite the shallowest water depth.*

**Answer:** We will modify the figure as requested.

**3_R#2:** *The agglutinated species composition (>150 microns) at site MC7 is similar to those of sites MC4 and MC5, but different from site MC6. The authors interpreted that the faunal similarity among sites might be caused by overflowing of the corrosive bottom water from the deep basin passing through the sills, but no explanations for the cause of faunal differences between site MC6 and others. (1) Is corrosive bottom water likely to overflow the basin without passing through site MC6, or flow out of the fjord through another path (e.g., channel)? (2) Another question arises about the possibility that basin species once carried outside the fjord have settled at the MC7 site. If so, a scenario overflowing from corrosive bottom water basins is no longer needed.*

**Answer:** As the results of the CCA (fig. 5) show, stations MC4, MC5 and MC6 are significantly similar, both in terms of fauna and environmental parameters. The agglutinated faunas of deep basins (MC4-MC5) and sill (MC6) stations only differ for the more abundant presence of *A. glomeratum* in this latter. We explain at lines 501-506 that the similarity with the deep basins fauna would be due to brine overflow, while the presence of *A. glomeratum* to the influence of the Atlantic Water especially during summer.

On the opposite, the fauna at station MC7 is significantly different from the others. The similarity between the agglutinated faunas of station MC7 and the deep basins and sill station is limited to the presence of *Reophax species*, but at this station the most abundant agglutinated species is *Lagenammina difflugiformis*, only marginally present in the deep basin stations. This species has been previously reported in wide environmental sets (e.g., Murray, 2006). It is often described as indifferent to organic matter supply (e.g., Alve et al., 2016; Jorissen et al., 2018) and sometimes has been reported in areas with hydro-sedimentary conditions characterized by more or less intermittent near-bottom currents (e.g., Fontanier et al., 2013). However, the most important difference concerning this station is determined by the prevalence of calcareous species and in particular the presence of two exclusive calcareous species: *Melonis barleanuum* and *Globobulimina auriculata*, indicating the strong influence of the Atlantic Water at this station.

With this in mind, we interpret the slightly different fauna of station MC6 as the result of intermittent pulses of BSW outflowing the fjord by-passing the sill during some periods of the year, and the presence of *A. glomeratum* as the evidence of seasonal influence of Atlantic waters.

We do not believe that the BSW outflow has a strong influence on station MC7. The presence of *Reophax* species at this station is more probably due to the widespread character of these agglutinated species (preference/tolerance for low quality organic matter). We do not think that a significant transport of individuals from the inner fjord to this station occurs; the common agglutinated species are quite big (up to 500µm!) and would be easily broken during transport. The silty nature of the sediments (average grain size value is about 14 µm) is clearly indicative of calm hydrodynamic environment at station MC7. Considering this, if transport happens via deep currents, it must be only minor. Moreover, no particular signs of reworking are visible on calcareous species and the same species are present in both large and small size fraction, comforting the absence of grain size sorting that could have happened via currents transport.

**14_R#2:** *The authors conclude that Agglutinated/Calcareous (A/C) proxies are possibly useful for changes in past fjord BSW intensity and sea ice production. However, the past A/C in sediments do not always reflect the marine environment at that time because agglutinated tests are more fragile than calcareous ones in general and are less likely to be preserved as fossils. How do the authors think about this?*

***Answer:*** it is true that agglutinated tests are generally not well preserved in the fossil record. However, Rasmussen & Thomsen (2014, 2015) conducted a study on long cores from the deep basin of Storfjorden dating back to 14kyr and observed that the preservation of agglutinated tests in these environments is particularly good. This is even more reliable for shorter preservation times, as we suggest the application of the A/C proxy on historical sedimentary records (as specified in lines 31-39 and 607-609).

**Minor comments**

- **R#2:** n text, references and captions "and" and "&" are mixed.

We will modify as requested.

- **R#2:** In text, figures, Tables and captions "subsp." is not required for "Elphidium excavatum subsp. clavatum".

We will modify as requested.

- **R#2:** line 88: Publication year of "Haarpaintner et al., 2001" should be 2001a, 2001b or 2001c.

There was a tiping error. For this sentence the only reference is Polyakov et al. 2012.

- **R#2:** line 89: "Polyakov et al., 2012" is missing in References.

We will add the reference as requested.

- **R#2:** line 108: "Fer, 2004" is missing in References.

We will add the reference as requested. Fer, I., Skogseth, R. and Haugan, P. M.: Mixing of the Storfjorden overflow (Svalbard Archipelago) inferred from density overturns, J. Geophys. Res., 109(C01005), doi:10.1029/2003JC001968, 2004.

- **R#2:** lines 109, 120: Publication year of "Skogseth et al., 2005" should be 2005a or 2005b.

We will correct with Skogseth et al., 2004

- **R#2:** line 204: "Pielou Index (1975)" needs author name(s) in the bracket.

We will modify as requested.

- **R#2:** line 434: "Rysgaard et al., 2011" is missing in References.

We will add the reference as requested. Rysgaard, S., Bendtsen, J., Delille, B., Dieckmann, G. S., Glud, R. N., Kennedy, H., Mortensen, J., Papadimitriou, S., Thomas, D. N. and Tison, J. L.: Sea ice contribution to the air-sea CO2 exchange in the Arctic and Southern Oceans, Tellus, Ser. B Chem. Phys. Meteorol., 63(5), 823–830, doi:10.1111/j.1600-0889.2011.00571.x, 2011.

- **R#2:** lines 536, 841: "Schroder-Adams" should be "Schröder-Adams".

We will modify as requested.

- **R#2:** line 540: "Jennings et Helgadottir, 1994 " should be "Jennings and Helgadottir, 1994 ".

We will modify as requested.

- **R#2:** line 662: Fer et al. (2004) is missing in the text.

The reference is cited at line 107. We will cite it again here, as suggested.

- **R#2:** line 683: Haarpaintner et al. (2001c) is missing in the text.

We will add this reference at line 68.

- **R#2:** line 704: Hunt and Corliss. (1993) is missing in the text.

We will modify as requested.

- **R#2:** line 750: Swap "Lloyd et al. (2007)" and "Lloyd (2006)".

We will modify as requested.

- **R#2:** lines 650, 660: Separate author's name with "and".

We will modify as requested.

- **R#2:** Figure 4: In legend "Adercotryma glomerata" should be "Adercotryma glomeratum", "Cribrostomoides crassimargo" should be "Labrospira crassimargo".

We will modify as requested.

- **R#2:** Figure S1: What does the difference in the color of the profile line at each sampling point indicate?

Different replicates of profiles. We will specify in the caption.

- **R#2:** Figure S4: For the taxonomy, SEM image of *Globocassidulina subglobosa* seems to be of *Cassidulina reniforme*.

We agree with the referee #2. We revised the taxonomy of those specimens and we assigned to all of them the name of *C. reniforme*. For that reason, we will modify the figure 6, 7 and 9 based on the new assignment of the all individuals to the same species. This modification does not change anything significant to the interpretation of data.

**References**

Alve, E., Korsun, S., Schönfeld, J., Dijkstra, N., Golikova, E., Hess, S., Husum, K. and Panieri, G.: Foram-AMBI: A sensitivity index based on benthic foraminiferal faunas from North-East Atlantic and Arctic fjords, continental shelves and slopes, Mar. Micropaleontol., 122, 1–12, doi:10.1016/j.marmicro.2015.11.001, 2016.

Fontanier, C., Metzger, E., Waelbroeck, C., Jouffreau, M., Lefloch, N., Jorissen, F., Etcheber, H., Bichon, S., Chabaud, G., Poirier, D., Grémare, A. and Deflandre, B.: Live (stained) benthic foraminifera off walvis bay, namibia: A deep-sea ecosystem under the influence of bottom nepheloid layers, J. Foraminifer. Res., 43(1), 55–71, doi:10.2113/gsjfr.43.1.55, 2013.

Jorissen, F., Nardelli, M. P., Almogi-Labin, A., Barras, C., Bergamin, L., Bicchi, E., El Kateb, A., Ferraro, L., McGann, M., Morigi, C., Romano, E., Sabbatini, A., Schweizer, M. and Spezzaferri, S.: Developing Foram-AMBI for biomonitoring in the Mediterranean: Species assignments to ecological categories, Mar. Micropaleontol., (January), doi:10.1016/j.marmicro.2017.12.006, 2018.

Murray, J. W.: Ecology and Application of Benthic Foraminifera, Cambridge University Press, Cambridge, New York., 2006.

Skogseth, R., Haugan, P. M. and Haarpaintner, J.: Ice and brine production in Storfjorden from four winters of satellite and in situ observations and modeling, J. Geophys. Res. C Ocean., 109(10), 1–15, doi:10.1029/2004JC002384, 2004.

---

## Author Comment (AC2) · 27 Jan 2020

**Major comments**

*1_R#1: The manuscript entitled 'Benthic foraminifera as tracers of brine production in Storfjorden "sea ice factory" by Eleanor Fossile and others definitely fits within the mission on Biogeosciences.*
*The hypothesis tested with the data presented is that the ratio of agglutinated to calcareous benthic foraminifera in Storfjorden, Svalbard archipelago, is largely controlled by brine formation and therefore can be used as a proxy for brine rejection processes and brine overflows in the paleo record. Previous research in the area has established that brine formation and overflow out of Storfjorden happen today and it has been inferred to have occurred in the past. The authors need to show that brine formation causes carbonate dissolution in modern samples and to rule out other processes such as high TOC causing low pH in the porewaters or show how other processes combine with the brine formation and overflow to cause dissolution.*

**Answer:** We thank the reviewer for the provided comments and suggestions that we will mostly follow in the revised version. Hereafter, we answer point-by-point the raised questions.

*2_R#1: I can see that it is important to have a proxy for the brine formation in this area and that it relates to the coastal polynya and the sea-ice factory, but please state clearly why it would be important for paleo studies to know if there were brines forming or not in the past. Why is your study significant? What does brine formation tell us about the sea ice conditions or climate/environment in the larger Arctic?*

**Answer:** Before answer the question we precise that our study proposes the A/C proxy for historical (e.g., 200-500 yrs) reconstruction of sea-ice production in the fjord, as stated in lines 31-39. We therefore concentrate on time scales much shorter than "paleo".
Studying brine evolution through recent time is important because brine circulation is a proxy for first-year sea-ice production and therefore for the functioning of the polynya system. This last has a crucial role in ocean circulation.
From Knies et al., 2017 "The Arctic Ocean halocline is maintained by the contribution of cold and brine-enriched deep waters (Aagaard et al., 1985; Cavalieri and Martin, 1994), which are formed because of high sea-ice production in coastal polynyas over the continental shelves (Fig. 7). Tamura and Ohshima (2011) showed that the current polar amplification of global warming will lead to negative trends in sea-ice production in most of the Arctic polynyas and with future projections of a summer ice-free Arctic Ocean (IPCC, 2013) sea-ice factories in Arctic coastal polynyas may lose their significance entirely."
The significance of these predictions, largely based on direct observations (i.e., satellite data of the last 50 years) needs to be evaluated on longer time scales, to place the recent trends in a longer-term perspective (i.e., multi-centennial time-scale) (Nicolle et al., 2018).
In our conclusion (lines 607-609) we already suggested the application of the A/C proxy on historical sedimentary records from Storfjorden in order to reconstruct recent changes in BSW intensity and, by extent, in sea ice production.
To satisfy the referee's questions, these different facets will be clarified in the introduction of the revised manuscript.

*3_R#1: You use a carefully developed dataset of living (biologically stained) foraminifera and environmental parameters such as various food types, hydrographic parameters and grain size from modern seabed samples to explore the ecology of the modern fauna. I find the paper to be carefully and clearly written in general. The development of the biozones and their association with various qualities of food source and hydrography is well done. However, I am not 100% convinced about the role of brine as the main driver of dissolution, but I think you could hone your arguments. I have some questions with that in mind.*
*Is there any chance of dissolution of calcareous faunas during your laboratory methods? Ethanol has a pH of 7.3. Did your samples sit in unbuffered water? You do not mention anything about buffering. Are there dead (unstained) calcareous forams in your samples? It seems like an important missing bit of information.*

**Answer:** We thank the referee for the appreciation of our work. About preservation, there are no reliable chances of dissolution related to our laboratory method. Ethanol is largely used as preservative method in the literature and recently suggested in the official protocol for living foraminiferal sample preservation by the FOBIMO group of specialists (Schönfeld et al., 2012). Moreover, Schönfeld et al. (2013) compared different

preservation methods for sediment samples (ethanol and formalin) and no signs of dissolution were observed in samples preserved with ethanol. Also, our samples never sit into unbuffered water during the processing.

Moreover, the preservation of the calcareous shells is much better in the inner fjord stations than in the deep basins, further suggesting that this result is not related to the preservation methods. In the inner fjord the dissolution only concerns some species, while others are not particularly affected (as stated in lines 397-402). Moreover, we observe preservation of calcareous species in the dead faunas, even if we decided not to include these data in the ms because they represent a too large dataset and only marginally useful for the purposes of the present ms. We will add some details about preservation methods in the "material and methods" chapter to clarify these points.

**4_R#1:** *Are there large variations in calcareous forams in the fossil record? Is that how people have inferred that there were brines in the past?*

**Answer:** We did not analyze proper fossil faunas, but only recently dead faunas (up to 6-8 cm depth in the sediment, corresponding to approximatively 40 years in the past; data not included in the ms). Of course the occurrence of taphonomical processes may affect differentially the difference between the dead and living faunas. In our study, the agglutinated fraction is relatively well preserved, and the A/C ratios applied on the dead faunas give coherent results.
As we report in the discussion (lines 540-545), Rasmussen and Thomsen, (2014, 2015) already observed shifts between agglutinated and calcareous ratios in the foraminiferal fossil record from the deep basin of the Storfjorden and suggested that it could reflect the intensification or weakening of brine productions during cold and warm periods respectively, during the last 14 kyr.

**5_R#1:** *The living fauna at the time of sample collection may not represent only this year or only one season, or it may exclude forams that bloomed earlier in the year. Give some insight into what the living fauna represents in terms of time. It does seem strange that living forams are badly dissolved! Are they really living at the time of collection or are they recently dead and already dissolving?*

**Answer:** As reported by Schönfeld et al. (2012) Rose Bengal stain is protein specific, and proteins are degraded fairly slowly under certain circumstances (e.g., hypoxic to anoxic conditions). Therefore, Rose Bengal may stain also proteins that are still in the shell after termination of metabolic activity, i.e., the death of the specimen (Bernhard, 1988; Murray and Bowser, 2000). In fact, the cytoplasm can be preserved in the test for some days to some weeks after the death of the foraminifera (Bernhard, 1988, 2000; Hannah and Rogerson, 1997; Murray and Bowser, 2000). Consequently, this method may lead to a slight overestimation of the living assemblages, especially if pale rose staining is considered enough to define vitality.
However, we are confident that the foraminiferal communities observed are representative of the late summer context of the fjord for three main reasons: 1) we only counted as living foraminifera with bright rose staining (assessing the coloration intensity of living specimens for every individual species, as recommended by Schönfeld et al., (2012)); it will be better described in the "materials and methods" chapter; 2) in well oxygenated environment, as our study area is, the residual cytoplasm after death would be quickly remineralised, reducing the possibilities of long term preservation for the necrotic cytoplasm; 3) we also report high abundances of juveniles of the same species found in the large size fraction, supporting the hypothesis that we sampled an active community.
This is not in contrast with the dissolution of calcareous species. In fact, there is a large literature reporting calcareous foraminifera able to survive with dissolved shells, both in natural and experimental setups. For example, Charrieau et al. (2017) report "zombies" foraminifera from the Swedish fjords, where pH was 7.4. Similarly, Pettit et al. (2013) report living (Cell-tracker green stained) foraminifera in environments with low pH (7.5). Bentov et al. (2009) were even able, in experimental conditions, to completely dissolve the calcareous test of a benthic foraminifer and let the specimen regrow a new test, confirming without doubts that the test dissolution is not inevitably cause of death.

**6_R#1:** *How do you determine what degree or type of staining points to a 'living' foram at the time of collection.*

**Answer:** The colour and intensity of Rose Bengal staining varies among species (Schönfeld et al., 2012). However a certain experience and a critical view minimize bias inferred by subjectivity (Murray and Bowser, 2000). According to the FOBIMO protocol we picked under wet conditions, which helps to preserve the

brightness of staining and we only counted bright rose colored specimens as alive. All doubtful staining were not taken into account. Some precisions about all that will be added in the revised ms.

**7_R#1:** *I suggest you add the word 'living' as a modifier of calcareous, agglutinated etc. more often because you are only presenting living assemblages and that really needs to be made clear. For example 4.5.1 Abundances and diversity of living forams.*

**Answer:** We think this is not necessary as the entire paper is about living foraminifera and we do not show any dead fauna data. However, we can accept this modification if also the editor thinks this is necessary.

**8_R#1:** *Can you provide a concise summary of why brine is corrosive to CaCO$_3$, along with explanation about how other factors (high CO$_2$, cold Arctic water and high TOC for example) interplay or potentially play their own role in the dissolution?*

**Answer:** There are several parameters to take into account to explain why brines are corrosive to CaCO$_3$. Calcite dissolution theoretically occurs when the carbon saturation state ($\Omega$Ca) is less than 1. The saturation point depends on pH, alkalinity, dissolved inorganic carbon, salinity and temperature. Brines form from Artic water, and are further enriched in CO$_2$ due to the rejection of inorganic carbon during the sea-ice production (Rysgaard et al., 2011, as reported in the chapter 2. "Oceanographic and environmental settings", lines 105-127). They are therefore necessarily richer in CO$_2$ compared to Arctic waters. Concerning the labile organic matter, its mineralization under oxic condition is indeed a cause of pH decrease, which is difficult to decouple from the pH decrease due to brines, as we stated several times in the ms (lines 441-444, 529-555). However, based on our data setwe can describe that the higher fresh and labile organic matter contents (the component which is mineralized the fastest), are recorded in the inner fjord stations, where the brine persistence is lesser and the calcareous faunas less affected in terms of dissolution. On the opposite, in the deep basins, where the brines persist all year round and where the organic matter is more refractory, the calcareous species are heavily dissolved. To us, this is a good argument to say that brines are most probably the main, even if surely not the only, controlling factor on calcareous test preservation.

We think this is already quite accurately explained in the presentation of the study area and in the discussions but we can do some slight modification to highlight these arguments in the discussion chapter of the revised ms.

**9_R#1:** *Storfjorden is called a fjord and you mention often about glacial meltwater and its influence on the headwaters of the fjord. But glaciers are not shown on your maps and I don't think they are described in your paper. That description is needed because you call on glacial meltwater and sediment delivery as an important part of the environmental gradient. On line 421 and 425 you use the term continental glacier, but I think you mean plateau ice cap or mountain glaciers?*

**Answer:** Spitsbergen is characterized by the presence of several tidewater glaciers (glaciers terminating with their calving front at the sea) influencing the head of the Storfjorden (see figure 9 in Lydersen et al., 2014). Many glaciers present on Svalbard retreated in the last 100 years because of climate warming increasing sediment supply and accumulation (Zajączkowski et al., 2004). This supply is in our opinion the source of terrigenous (refractory) organic carbon in the fjord.
With continental glaciers we meant tidewater glaciers. We will change the term in the manuscript. These glaciers are present all around the fjord (we may write in the "study area" chapter to refer to figure 9 in Lydersen et al., (2014) for more details on glacier distribution), but we do not think that it is important for the purpose of our manuscript to show them on the map as we do not identify any of them in particular as direct responsible for sediment supply.

**10_R#1:** *And Storfjorden really looks like a sound as it forms a connection between the Barents and Greenland seas via Heleysundet and Freemansundet. What role do these connections play in the fjord hydrography? How important is the ESC waters that come into the head of the fjord for the formation of brines? Can the differences you see in MC3s be related to its proximity to Freemansundet? Your map figure is so small that I could not easily read the labels.*

**Answer:** Indeed, the Storfjorden makes a connection with the Northern Barents Sea via Heleysundet and Freemansundet. We explained the role of these sounds at lines 73-75, 94-96. The ESC current carries cold Arctic waters, contributing to maintain an active cyclonic circulation. These waters in winter-early spring, thanks to the katabatic winds enhancing the polynya, contribute to the formation of BSW (Skogseth et al.,

2005). The figure was maybe reduced in size during the formatting processes (?). We will take care that the labels are clearly visible in the final version.

**11_R#1:** *On Figure 1b, add the Atlantic Water..you can use a special arrow or something. Also add the Arctic water. Can you add the polynya to Figure 1a?*

**Answer**: Atlantic waters are already present on the figure 1a. As stated in the figure caption, they are represented by the red arrows and by the Norwegian Atlantic Current (NAC) and the West Spitsbergen Current (WSC). We prefer not to add the polynya to the map because its extension is hugely variable (interannually and annually). Haarpaintner et al., (2001) clearly explain the variability of the polynya and propose some figures to show that. We will add a short statement to provide this information.

**12_R#1:** *On page 4 you discuss the organic matter composition of the sediments and the potential of a terrestrial component. What is the bedrock geology of this area? Can some refractory carbon be from bedrock erosion and deposition in the fjord?*

**Answer:** The organic matter we are talking about is the one mainly coming from the continental soil and not from the bedrock. This organic matter is eroded and drained into the fjord especially during the melting season. Please refer to the answer concerning the tidewater glaciers above (comment 9).

**13_R#1:** *Line 125. How does Storfjorden introduce brine to the Arctic Ocean when it drains to the Greenland Sea?*

**Answer:** As represented in figure 1, the ESC (blue arrow), enters the Storfjorden, where it is enriched of BSW, then outflows, canalized by the Polar Front, along the western coast of Spitsbergen towards the Arctic Ocean.

**13_R#1:** *Line 545. Explain what it is about brines that make them corrosive to carbonate. You have said they have high $CO_2$ content, which is also what the Arctic Surface waters have that enter the fjord. I think you are getting at several factors that converge to make acidic pore waters in the fjord basins one of which is brines. Clarify and organize this argument.*

**Answer:** please see the answer to the comment 8.

**14_R#1** *In your conclusions you also mash together the brine and other factors that can cause dissolution together (Lines 595-597) but the takeaway is that the dissolution is because of the brine formation. Can you clarify this and maybe state that brines are associated with some other conditions that converge to cause dissolution?*

**Answer:** we already answer this comment above (see comment 8). We will further clarify in the revised version that we cannot exclude the role of other factors (such as organic matter remineralization) on calcareous test dissolution, but we have quite strong argument to say that brines certainly play a major role.

**Minor comments**

- **R#1:** *45 in the meantime*

We will modify as requested.

- **R#1:** *105 clarify this sentence. If there is a persistent polynya then why is there extended winter sea ice cover as well?*

We will modify winter with first year sea-ice.

- **R#1:** *138 (10 cm diameter)*

We will modify as requested.

- **R#1:** *150 microelectrode*

We accept the modification requested.

- **R#1:** *162 'replicate analyses' Not replicated? I find this sentence unclear. How do you know which sample is most representative?*

We will modify as requested. Since the machine measures three times the same samples and we measured two replicates for each layer, at the end we have 6 measures of the same sample. In our opinion, it is not correct to make a mean of the six samples; it is better to select the measure which is the best compromise among the six.

- **R#1:** *166 Pb dating was. . .*

We will modify as requested.

- **R#1:** *184 30 µg C µg phytopigment -1. Is this the correct way to state this? It is awkward*

We can say 30 $\mu$g C $\mu$g per phytopigment.

- **R#1:** *247 describe the silt % in various samples and its range in percent. Say that 20µm is medium silt and 10 µm is fine silt.*

Slight differences are however noted in terms of the mode (approximately 10 µm, fine silt, in the fjord and 20 µm, medium silt, at the outer station MC7) and the percentage of sand which increases from approximately 4% at MC1 to 10.4% at station MC6.

- **R#1:** *249 and declines to 6.8% at MC7*

We will modify as requested.

- **R#1:** *258 not lower that (n). . .have to say 'less than or equal to'. . .*

The inner fjord stations (MC1-MC3) present $pH_T$ values generally above 7.95 whereas the deep basin stations display values less than 7.90 units (7.84 and 7.90 for MC4 and MC5 respectively).

- **R#1:** *297 Elphidium clavatum is considered to be a separate species now. See Darling et al., 2016 in Marine Micropaleo v. 129, p 1-23.*

We will modify the name in the entire text.

- **R#1:** *P.10 suggest you add the word living to modify foraminifera in this section. You need not do this every time, but use this modifier in the top of each section and especially in the heading so that it is clear that your total assemblage is limited to living fauna.*

If it is no too redundant for the editor, we will do this modification.

- **R#1:** *354 change 'distinguishes also to separate' to distinguishes*

We will modify as requested.

- **R#1:** *477 italicize E.*

We will modify as requested.

- **R#1:** *478 Melonis has been associated with degraded OM (Caralp, 1989)*

Caralp, 1989 states "A high percentage of *Melonis barleeanum* in deep-sea benthic foraminiferal assemblages is related to the availability of food in the form of abundant, little-altered, marine organic matter." We already wrote (line 511)": In the Atlantic Ocean, this species is described as opportunist in response to good quality organic matter (e.g., Nardelli et al., 2010). We will also add the citation of Caralp (1989), that supports the same.

- **R#1:** *479 G. auriculata is often associated with buried OM*

We will add this information about *G. auricolata*.

- ***R#1:*** *533 I don't know if it is true that the most obvious explanation for the severe dissolution is the brine. You really need to build this argument. This lack of building an argument about the affect of brine and the other factors associated with Arctic water and TOC weakens the paper.*

Please refer to answer 8 in the major comments

- ***R#1:*** *540 change et to and*

We will modify as requested.

***R#1:*** *Figures: The stacked histograms showing species at sites are really hard to read. A major problem is that the key is so small that a person cannot see the pattern. I like the idea, but it may be better to make histograms of species in each site and stack them one above the other. Or choose fewer species. For example, you could use only the species found to be statistically significant*

We prepared a bigger legend to make the textures of the legend more visible. We think that the pdf version corrected by the reviewer probably lost a bit of quality and we are sure that the texture will be understandable once the high definitions figures will be used.

**References**

Aagaard, K., Swift, J. H. and Carmack, E. C.: Thermohaline circulation in the Arctic Mediterranean Seas, J. Geophys. Res., 90(C3), 4833, doi:10.1029/JC090iC03p04833, 1985.

Bentov, S., Brownlee, C. and Erez, J.: The role of seawater endocytosis in the biomineralization process in calcareous foraminifera, Proc. Natl. Acad. Sci. U. S. A., 106(51), 21500–21504, doi:10.1073/pnas.0906636106, 2009.

Bernhard, J. M.: Postmortem vital staining in benthic foraminifera; duration and importance in population and distributional studies, J. Foraminifer. Res., 18(2), 143–146, doi:10.2113/gsjfr.18.2.143, 1988.

Bernhard, J. M.: Distinguishing Live from Dead Foraminifera : Methods Review and Proper Applications, micropaleontolgy Proj. Inc., 46(1), 38–46, 2000.

Cavalieri, D. J. and Martin, S.: The contribution of Alaskan, Siberian, and Canadian coastal polynyas to the cold halocline layer of the Arctic Ocean, J. Geophys. Res., 99(C9), 18,343-18,362, doi:0148-0227/94/94JC-01169505.00, 1994.

Charrieau, L. M., Filipsson, H. L., Nagai, Y., Kawada, S., Ljung, K., Kritzberg, E. and Toyofuku, T.: Decalcification and survival of benthic foraminifera under the combined impacts of varying pH and salinity, Lundqua Thesis, 2017(March), 85–99, doi:10.1016/j.marenvres.2018.03.015, 2017a.

Charrieau, L. M., Filipsson, H. L., Ljung, K., Chierici, M., Knudsen, K. L. and Kritzberg, E.: The effects of multiple stressors on the distribution of coastal benthic foraminifera: A case study from the Skagerrak-Baltic sea region, Lundqua Thesis, 2017(November 2017), 57–81, doi:10.1016/j.marmicro.2017.11.004, 2017b.

Haarpaintner, J., Haugan, P. M. and Gascard, J. C.: Interannual variability of the Storfjorden (Svalbard) ice cover and ice production observed by ERS-2 SAR, Ann. Glaciol., 33, 430–436, 2001.

Hannah, F. and Rogerson, A.: The temporal and spatial distribution of foraminiferans in marine benthic sediments of the Clyde Sea Area, Scotland, Estuar. Coast. Shelf Sci., 44(3), 377–383, doi:10.1006/ecss.1996.0136, 1997.

IPCC: Climate Change 2013: The Physical Science Basis. Working Group I Contribution to the Fifth Assessment Report of the Intergovernmental Panel on Climate Change [Stocker, T.F., D. Qin, G.-K. Plattner, M. Tignor, S.K. Allen, J. Boschung, A. Nauels, Y. Xia, V., Cambridge Univ. Press. Cambridge, United Kingdom New York, NY, USA, 1535 pp., doi:10.1017/CBO9781107415324, 2013.

Knies, J., Pathirana, I., Cabedo-Sanz, P., Banica, A., Fabian, K., Rasmussen, T. L., Forwick, M. and Belt, S. T.: Sea-ice dynamics in an Arctic coastal polynya during the past 6500 years, Arktos, 3(1), 1, doi:10.1007/s41063-016-0027-y, 2017.

Lydersen, C., Assmy, P., Falk-Petersen, S., Kohler, J., Kovacs, K. M., Reigstad, M., Steen, H., Strøm, H., Sundfjord, A., Varpe, Ø., Walczowski, W., Weslawski, J. M. and Zajaczkowski, M.: The importance of tidewater glaciers for marine mammals and seabirds in Svalbard, Norway, J. Mar. Syst., 129, 452–471, doi:10.1016/j.jmarsys.2013.09.006, 2014.

Murray, J. W. and Bowser, S. S.: Mortality, protoplasm decay rate, and reliability of staining techniques to recognize "living" foraminifera: A review, J. Foraminifer. Res., 30(1), 66–70, doi:10.2113/0300066, 2000.

Nicolle, M., Debret, M., Massei, N., Colin, C., Devernal, A., Divine, D., Werner, J. P., Hormes, A., Korhola, A. and Linderholm, H. W.: Climate variability in the subarctic area for the last 2 millennia, Clim. Past, 14(1), 101–116,

doi:10.5194/cp-14-101-2018, 2018.

Pettit, L. R., Hart, M. B., Medina-Sánchez, A. N., Smart, C. W., Rodolfo-Metalpa, R., Hall-Spencer, J. M. and Prol-Ledesma, R. M.: Benthic foraminifera show some resilience to ocean acidification in the northern Gulf of California, Mexico, Mar. Pollut. Bull., 73(2), 452–462, doi:10.1016/j.marpolbul.2013.02.011, 2013.

Rasmussen, T. L. and Thomsen, E.: Brine formation in relation to climate changes and ice retreat during the last 15,000 years in Storfjorden, Svalbard, 76 – 78°N, Paleoceanography, 911–929, doi:10.1002/2014PA002643.Received, 2014.

Rasmussen, T. L. and Thomsen, E.: Palaeoceanographic development in Storfjorden, Svalbard, during the deglaciation and Holocene: Evidence from benthic foraminiferal records, Boreas, 44(1), 24–44, doi:10.1111/bor.12098, 2015.

Rysgaard, S., Bendtsen, J., Delille, B., Dieckmann, G. S., Glud, R. N., Kennedy, H., Mortensen, J., Papadimitriou, S., Thomas, D. N. and Tison, J. L.: Sea ice contribution to the air-sea CO2 exchange in the Arctic and Southern Oceans, Tellus, Ser. B Chem. Phys. Meteorol., 63(5), 823–830, doi:10.1111/j.1600-0889.2011.00571.x, 2011.

Schönfeld, J., Alve, E., Geslin, E., Jorissen, F., Korsun, S., Spezzaferri, S., Abramovich, S., Almogi-Labin, A., du Chatelet, E. A., Barras, C., Bergamin, L., Bicchi, E., Bouchet, V., Cearreta, A., Di Bella, L., Dijkstra, N., Disaro, S. T., Ferraro, L., Frontalini, F., Gennari, G., Golikova, E., Haynert, K., Hess, S., Husum, K., Martins, V., McGann, M., Oron, S., Romano, E., Sousa, S. M. and Tsujimoto, A.: The FOBIMO (FOraminiferal BIo-MOnitoring) initiative-Towards a standardised protocol for soft-bottom benthic foraminiferal monitoring studies, Mar. Micropaleontol., 94–95, 1–13, doi:10.1016/j.marmicro.2012.06.001, 2012.

Schönfeld, J., Golikova, E., Korsun, S. and Spezzaferri, S.: The Helgoland Experiment – assessing the influence of methodologies on Recent benthic foraminiferal assemblage composition, J. Micropalaeontology, 32(2), 161–182, doi:10.1144/jmpaleo2012-022, 2013.

Skogseth, R., Fer, I. and Haugan, P. M.: Dense-water production and overflow from an arctic coastal polynya in storfjorden, Geophys. Monogr. Ser., 158, 73–88, doi:10.1029/158GM07, 2005.

Tamura, T. and Ohshima, K. I.: Mapping of sea ice production in the Arctic coastal polynyas, J. Geophys. Res. Ocean., 116(7), 1–20, doi:10.1029/2010JC006586, 2011.

Zajączkowski, M., Szczuciński, W. and Bojanowski, R.: Recent changes in sediment accumulation rates in Adventfjorden, Svalbard, Oceanologia, 46(2), 217–231, 2004.